# Microcultivation and FTIR spectroscopy-based screening revealed a nutrient-induced co-production of high-value metabolites in oleaginous *Mucoromycota* fungi

**Simona Dzurendova**[1]*, **Boris Zimmermann**[1], **Achim Kohler**[1], **Valeria Tafintseva**[1], **Ondrej Slany**[2], **Milan Certik**[2], **Volha Shapaval**[1]

**1** Faculty of Science and Technology, Norwegian University of Life Sciences, Ås, Norway, **2** Faculty of Chemical and Food Technology, Slovak Technical University, Bratislava, Slovakia

* simona.dzurendova@gmail.com, simona.dzurendova@nmbu.no

**Data Availability Statement:** All relevant data are within the paper and its Supporting Information files.

## Abstract

*Mucoromycota* fungi possess a versatile metabolism and can utilize various substrates for production of industrially important products, such as lipids, chitin/chitosan, polyphosphates, pigments, alcohols and organic acids. However, as far as commercialisation is concerned, establishing industrial biotechnological processes based on *Mucoromycota* fungi is still challenging due to the high production costs compared to the final product value. Therefore, the development of co-production concept is highly desired since more than one valuable product could be produced at the time and the process has a potentially higher viability. To develop such biotechnological strategy, we applied a high throughput approach consisting of micro-titre cultivation and FTIR spectroscopy. This approach allows single-step biochemical fingerprinting of either fungal biomass or growth media without tedious extraction of metabolites. The influence of two types of nitrogen sources and different levels of inorganic phosphorus on the co-production of lipids, chitin/chitosan and polyphosphates for nine different oleaginous *Mucoromycota* fungi was evaluated. FTIR analysis of biochemical composition of *Mucoromycota* fungi and biomass yield showed that variation in inorganic phosphorus had higher effect when inorganic nitrogen source–ammonium sulphate–was used. It was observed that: (1) *Umbelopsis vinacea* reached almost double biomass yield compared to other strains when yeast extract was used as nitrogen source while phosphorus limitation had little effect on the biomass yield; (2) *Mucor circinelloides*, *Rhizopus stolonifer*, *Amylomyces rouxii*, *Absidia glauca* and *Lichtheimia corymbifera* overproduced chitin/chitosan under the low pH caused by the limitation of inorganic phosphorus; (3) *Mucor circinelloides*, *Amylomyces rouxii*, *Rhizopus stolonifer* and *Absidia glauca* were able to store polyphosphates in addition to lipids when high concentration of inorganic phosphorus was used; (4) the biomass and lipid yield of high-value lipid producers *Mortierella alpina* and *Mortierella hyalina* were significantly increased when high concentrations of inorganic phosphorus were combined with ammonium sulphate, while the same amount of inorganic phosphorus combined with yeast extract showed negative impact on the growth and lipid accumulation. FTIR spectroscopy revealed the co-production potential of several

**Funding:** The study was funded by the Research Council of Norway - FMETEKN Grant, project number 257622; BIONÆR Grant, project numbers 268305, 305215; DAAD Grant, project number 309220 and by the Slovak Ministry of Education, Science, Research and Sport- grant VEGA 1/0323/19. The funders had no role in study design, data collection and analysis, decision to publish, or preparation of the manuscript.

**Competing interests:** The authors have declared that no competing interests exist.

oleaginous *Mucoromycota* fungi forming lipids, chitin/chitosan and polyphosphates in a single cultivation process.

## Introduction

Biorefinery is the sustainable processing of biomass into a spectrum of marketable products, such as biofuels and biochemicals, through the application of green conversion technologies [1]. *Mucoromycota* filamentous fungi play an important role in developing sustainable biorefinery processes due to their versatile metabolism and ability to utilize a broad range of renewable feedstock, rest and waste materials [2–4]. *Mucoromycota* fungi are able to produce a number of industrially important products, such as alcohols, organic acids and enzymes [5]. Moreover, the biomass of *Mucoromycota* fungi is rich in various high-value metabolites such as lipids, proteins, pigments, polyphosphates and chitosan [6], making it well suited for nutrition purposes as a whole.

It is well known that some filamentous fungi, so-called oleaginous fungi, are able to produce high amounts of lipids. Oleaginous *Mucoromycota* fungi are able to accumulate lipids (Single Cell Oils–SCOs) with up to 80% w/w yield [7]. SCOs are stored in globular intracellular organelles (i.e. lipid bodies) predominantly in the form of triacylglycerides (TAGs) [8, 9]. Depending on a fungal strain, fungal lipids can be very similar to vegetable oils and thus suitable for biodiesel production, or similar to highly nutritious and valuable fish oils with high content of polyunsaturated fatty acids (PUFAs) [10–13]. Although SCOs production by oleaginous *Mucoromycota* fungi has been suggested and up-scaled a century ago [14], industrial process based on *Mucoromycota* fungi are still limited only to production of high-value PUFA-rich oils. For example gamma-linoleic and arachidonic acids rich oils are produced industrially by *Mucor circinelloides* and *Mortierella alpina* [15].

Despite all the developments in the field of fungal SCOs, a commercially sustainable *Mucoromycota*-based biodiesel production has not yet been established. The production of relatively low-valued fungal lipids, such as biodiesels, could become economically feasible if a concept of co-production of lipids and other value-added chemicals is applied [16]. Such co-production concept for oleaginous *Mucoromycota* fungi has been first suggested for production of chitosan and biodiesel lipids by *Mucor circinelloides* [17]. Other co-production strategies using *Mucoromycota* include concomitant production of fumaric acid and chitin by *Rhizopus oryzae* [18], lactic acid and chitin by *Rhizopus oryzae* [19], lipids, proteins, ethanol and chitosan by *Rhizopus oryzae* and *Mucor indicus* [20].

When developing a sustainable co-production strategy for oleaginous *Mucoromycota* fungi, the aim is to co-produce high-value metabolites which are generated in metabolic pathways that are not competing for carbon sources. Thus, co-production of extracellular acids and intracellular lipids is expected to have low sustainability since these metabolic pathways are competing for the carbon source. Notwithstanding, strategies based on the co-production of metabolites of lipid bodies and cell wall are advantageous since metabolic pathways for the production of chemical components of these two organelles are not competing. The major components of the cell wall in *Mucoromycota* fungi are commercially lucrative biopolymers chitin, chitosan and polyphosphates [21].

Chitin (ß-1,4-N-acetyl-D-glucosamine) and its deacetylated form, chitosan (ß-1,4-D-glucosamine), are natural biodegradable polymers with a broad range of applications in food, pharmaceutical and agricultural industries [22]. Chitosan belongs to the most versatile and

promising functional biopolymers, with superior material properties and interesting biological activities. An increasing market demand for high-quality chitosan exceeds the current global production, which is based primarily on deacetylation of chitin from shells of crustaceans. Therefore, production based on *Mucoromycota* fungi could be lucrative, in particular since *Mucoromycota* are among the rather rare natural producers of chitosan. In some cases, the total chitin and chitosan yield in *Mucoromycota* fungi can reach up to 40% w/w [23].

Another important biopolymer of the *Mucoromycota* cell wall is polyphosphate [24], a chain of phosphate units connected by high-energy phospho-anhydride bonds. Polyphosphates have several key functions in fungal cells, such as energy and phosphate storage, controlling of fungal homeostasis via trapping cations and amino acids, and regulation of the hyphal phosphate amount [25]. Phosphorus accumulation takes place in the exponential growth phase, when the source of phosphorus is in high access [26]. Phosphorus accumulating *Mucoromycota* fungi are able to store more polyphosphates than needed for their survival, which is very attractive for the phosphorus recovery. Currently, the global phosphorus market is getting into a critical situation due to the limited availability of rock phosphate, which is a non-renewable phosphorus source. Various waste substrates contain significant amounts of phosphorus that could be recovered if appropriate processes for phosphorus recovery were available. Waste sources of phosphorus are municipal waste or waste-water streams [27]. The traditional phosphorus recovery approach is based on wet-chemistry and thermo-chemical treatment. It requires the use of chemicals and high energy [28]. An alternative and more sustainable way of phosphorus recovery is based on utilizing filamentous fungi that are able to accumulate phosphorus during their growth [26]. Therefore, production of fungal polyphosphate in a biorefinery concept can have a significant contribution to phosphorus recycling. However, not all fungal strains possess the ability to accumulate phosphorus and therefore biotechnologically valuable phosphorus-accumulating strains need to be identified.

The development of sustainable fungal biorefinery for co-production of lipids, chitin/chitosan and polyphosphates depends sensitively on the chemical composition of the substrates since different sources are required for the different metabolic processes needed to reach the target products. When processes are built on the utilization of different rest materials and waste streams as low-cost substrates, theses substrates need to be modified and optimized such that they contain all needed sources in the best possible concentrations. Rest materials and waste streams have a highly diverse chemical composition and there is a need to enrich them with essential macro- and micro-nutrients. Therefore, the optimization of cultivation media or substrates based on rest materials and waste streams is crucial in the fungal biorefinery process development. In order to perform adequate optimizing of rest materials-based substrates, there is a need for deeper understanding the effect of single media components on the synthesis of different intra- and extracellular metabolites in fungal cells. Extensive research has been done on studying the role of different carbon and to some extend nitrogen sources [29–33] for the fungal fermentation in general and for the production of one main metabolite. However, there is a very little known about the role of phosphorus on the production of different metabolites, as this element is mostly examined in the context of polyphosphates accumulation [34]. In addition, to our knowledge no study has been performed so far that investigates the effect of single media nutrients either in excess or in limited amounts on the co-production of several metabolites by fungi.

The traditional approach for monitoring and developing the production of different metabolites in fungal cells is based on the extraction or separation of the produced metabolites followed by further qualitative and quantitative analysis using different analytical procedures. Such approach requires significant amount of biomass for the analysis, since different metabolites need to be extracted and analyzed in different and often expensive and time-consuming ways. Fourier Transform Infrared (FTIR) spectroscopy is a rapid non-invasive technology

allowing biochemical fingerprinting of all cell chemical components [35]. While FTIR spectroscopy has been used for many decades for structural chemical analysis, FTIR spectroscopy became a popular tool for identification and characterization of biological materials in the 90ies. FTIR spectroscopy has been extensively used in applied microbiology and biotechnology of various types of microorganisms, including fungi [36–41], bacteria [42–44], yeasts [45–50] or algae [51–53]. Moreover, FTIR spectroscopy was applied as a tool for measurement of growth media and extracellular metabolites [54, 55]. FTIR spectroscopy has been shown to be precise and reliable method for the identification and analysis of microbial lipids [10, 56–61], chitin/chitosan [62–65] and polyphosphates [66, 67]. Further, this method has been utilized for monitoring lipid extraction in oleaginous filamentous fungi [68, 69]. Thus, FTIR showed the potential to serve as the sole method for the bioprocess monitoring. Combined with the Duetz microtiter plate system (Duetz-MTPS), FTIR can serve as a rapid tool for monitoring of high throughput studies, such as screening of fungal strains for high and low value lipid production [10, 70]. Moreover, high throughput screening was strengthen by a fully automated set-up of the biomass samples preparation for the FTIR-HTS analysis [71–73].

The aim of this study was to assess the biotechnological potential of oleaginous *Mucoromycota* grown on two different nitrogen sources, namely yeast extract and ammonium sulphate, in combination with six different inorganic phosphorus (Pi) concentrations in a high throughput screening using FTIR spectroscopy combined with Duetz-MTPS. The primary goal of presented high throughput screening is the identification of co-producing strains and understanding the role of phosphorus and nitrogen alone and in the interaction in the co-production. Thus, the study provides relative estimation of the high-value metabolites co-produced by *Mucoromycota* fungi and, therefore, can be considered as a basis for further research in developing of co-production concepts.

## 1. Materials and methods

### 1.1. Oleaginous filamentous fungi

Nine oleaginous filamentous fungi from the genera *Absidia*, *Amylomyces*, *Cunninghamella*, *Lichthemia*, *Mortierella*, *Mucor*, *Rhizopus* and *Umbelopsis* were used in the study (Table 1). The selection of fungal strains was based on the results of our recent study, where 100 oleaginous filamentous fungi were screened for the ability to accumulate high amount of lipids [10].

**Table 1. List of oleaginous filamentous fungi used in the study.**

| Fungal strain name | Collection № | Short name |
|---|---|---|
| *Absidia glauca* | CCM[1] 451 | AGL |
| *Amylomyces rouxii* | CCM F220 | ARO |
| *Cunninghamella blakesleeana* | CCM F705 | CBL |
| *Lichtheimia corymbifera* | CCM 8077 | LCO |
| *Mortierella alpina* | ATCC[2] 32222 | MAL |
| *Mortierella hyalina* | VKM[3] F1629 | MHY |
| *Mucor circinelloides* | VI[4] 04473 | MCI |
| *Rhizopus stolonifer* | VKM F-400 | RST |
| *Umbelopsis vinacea* | CCM F539 | UVI |

[1]Czech collection of Microorganisms (Brno, Czech Republic)

[2]American Type Culture Collection (Virginia, USA)

[3]All-Russian Collection of Microorganisms (Moscow, Russia), and

[4]Norwegian school of Veterinary Science (Oslo, Norway).

While some Mucoromycota species have been previously identified as medically important [74], in general they have been utilised at industrial scale as cell factories for example for chitosan, lipids or lactic acid production.

## 1.2. Design of the experiment

Six different concentrations of phosphate salts–$KH_2PO_4$ and $Na_2HPO_4$, and two different nitrogen sources–yeast extract (YE) and ammonium sulphate (AS)–were used for the cultivation of fungi in a full factorial design. The cultivation was performed in Duetz-MTPS [54] in three independent biological replicates for each fungus, phosphorus concentration and nitrogen source, resulting in 324 samples. Biological replicates were prepared on separate microtiter plates and cultivated at different time points for each fungal strain. For every biological replicate, fresh spore suspension was prepared.

## 1.3. Growth media and cultivation conditions

Growth of the selected fungi was done in two steps: 1) growth on standard agar medium for preparing spore inoculum and 2) growth in nitrogen-limited broth media with different inorganic phosphorus (Pi) concentrations and nitrogen sources in the Duetz-MTPS.

For the preparation of spore inoculum, *Mortierella* and *Umbelopsis* were cultivated on potato dextrose agar, while all other strains were cultivated on malt extract agar. Malt extract agar was prepared by dissolving 30 g of malt extract (Merck, Germany), 5 g of peptone (Amresco, USA) and 15 g of agar powder (Alfa Aesar, ThermoFischer, Germany) in 1L of distilled water and autoclaved at 115˚C for 15 min. Potato dextrose agar was prepared by dissolving 39 g of potato dextrose agar (VWR, Belgium) in 1L of distilled water and autoclaved at 115˚C for 15 min. Agar cultivation was performed for 7 days at 25˚C for all strains except for *Mortierella* (14 days) due to the slower growth of *Mortierella*. Fungal spores were harvested from agar plates with a bacteriological loop after the addition of 10 mL of sterile 0.9% NaCl solution.

The main components of the nitrogen-limited broth media [75] with modifications [76] (g·$L^{-1}$) were: glucose 80, yeast extract 3, MgSO4·7H2O 1.5, CaCl2·2H2O 0.1, FeCl3·6H2O 0.008, ZnSO4·7H2O 0.001, CoSO4·7H2O 0.0001, CuSO4·5H2O 0.0001, MnSO4·5H2O 0.0001. For broth media with ammonium sulphate as a nitrogen source, yeast extract was replaced with 1.5 g/L of $(NH_4)_2SO_4$ in order to keep the same C/N ratio as with yeast extract medium. Broth media with ammonium sulphate contained 0.05g/L thiamin hydrochloride and 0.02 mg/L biotin [77]. Different concentrations of phosphate salts, namely $KH_2PO_4$ and $Na_2HPO_4$, were added to the main components of nitrogen-limited broth medium, as described in Table 2. The concentrations of phosphate salts, 7 g·$L^{-1}$ $KH_2PO_4$ and 2 g·$L^{-1}$ $Na_2HPO_4$, were selected as a reference value (Pi1) since they have frequently been used in cultivation of oleaginous *Mucoromycota* [75, 76]. The broth media contained higher (up to 8 × Pi1) and lower (up to ¼ × Pi1) amount of phosphate salts compared to the reference value (Table 2). Broth media with

**Table 2. The concentration of phosphate salts in the nitrogen-limited broth media.**

| Sample name | $KH_2PO_4$ (g·$L^{-1}$) | $Na_2HPO_4$ (g·$L^{-1}$) |
|---|---|---|
| Pi8 | 56 | 16 |
| Pi4 | 28 | 8 |
| Pi2 | 14 | 4 |
| Pi1 | 7 | 2 |
| Pi0.5 | 3.5 | 1 |
| Pi0.25 | 1.75 | 0.5 |

the decreased amount of inorganic phosphorus contained KCl and NaCl in a corresponding concentration in order to have equal K$^+$ and Na$^+$ ions as in the reference condition (Pi1). Broth media were autoclaved for 15 min at 121˚C. The starting pH of media was 6 ± 0.3 and pH of growth media was measured as well after the cultivation.

Cultivation in broth media was performed in the Duetz-MTPS (Enzyscreen, Netherlands) which consists of 24-square polypropylene deep well microtiter plates, low evaporation sandwich covers and extra high cover clamps, which were placed into the MAXQ 4000 shaker (Thermo Scientific). The autoclaved microtiter plates were filled with 7 ml of sterile broth media per well, and each well was inoculated with 50 μl of spore inoculum. Cultivation was performed for 7 days at 25˚C and 400 rpm agitation (1.9 cm circular orbit). Fungi *Mortierella alpina* and *Mortierella hyalina* were cultivated for 14 days due to their slow growth.

### 1.4. Analysis

**1.4.1. Fourier transform infrared spectroscopy of fungal biomass.** Fourier transform infrared (FTIR) spectroscopy analysis of fungal biomass was performed according to [76] with some modifications. The biomass was separated from the growth media by centrifugation and washed with distilled water. Approximately 5 mg of fresh washed biomass was transferred into 2 ml polypropylene tube containing 250±30 mg of acid washed glass beads and 0.5 ml of distilled water for further homogenization. The remaining washed biomass was freeze-dried for 24 hours for determining biomass yield. In total, 312 biomass samples were analysed in three technical replicates by FTIR spectroscopy. *Mortierella alpina* and *Mortierella hyalina* in Pi8 conditions were not measured since no growth was observed.

The homogenization of fungal biomass was performed by using Percellys Evolution tissue homogenizer (Bertin Technologies, France) with the following set-up: 5500 rpm, $6 \times 20$ s cycle. 10 μl of homogenized fungal biomass was pipetted onto an IR transparent 384-well silica microplate. Samples were dried at room temperature for two hours. For every sample, three technical replicates were prepared. The FTIR spectra were recorded in the region between 4000 cm$^{-1}$ and 500 cm$^{-1}$ with a spectral resolution of 6 cm$^{-1}$, a digital spacing of 1.928 cm$^{-1}$, and an aperture of 5 mm. Spectra were recorded in a transmission mode using the High Throughput Screening eXTension (HTS-XT) unit coupled to the Vertex 70 FTIR spectrometer (both Bruker Optik, Germany). For each spectrum, 64 scans were averaged. Spectra were recorded as the ratio of the sample spectrum to the spectrum of the empty IR transparent microplate. In total, 936 biomass spectra were obtained. The OPUS software (Bruker Optik GmbH, Germany) was used for data acquisition and instrument control.

**1.4.2. Attenuated total reflectance Fourier transform infrared spectroscopy.** Attenuated total reflectance (ATR)-infrared spectra of growth media after cultivation, were recorded using a Vertex 70 FTIR spectrometer (Bruker Optik GmbH, Germany) with a single reflectance-attenuated total-reflectance accessory. For identification of basic biochemicals in the growth media and the biomass a set of model compounds was measured. Spectra of glyceryl trioleate ((9Z)9-Octadecenoic acid 1,2,3-propanetriyl ester), chitin, and sodium polyphosphate were measured. Moreover, spectra of pure water, and water solutions of pure glucose, ammonium sulphate, yeast extract, phosphates, as well as media before cultivation were recorded. All chemicals were purchased from Merck (Darmstadt, Germany) and used without further purification. In addition to growth media, spectra of pure water, and water solutions of pure glucose, ammonium sulphate, yeast extract, phosphates, as well as media before cultivation were recorded. The ATR IR spectra were recorded with a total of 32 scans and spectral resolution of 4 cm$^{-1}$ over the range of 4000–600 cm$^{-1}$, using the horizontal ATR diamond prism with 45˚ angle of incidence on a High Temperature Golden gate ATR Mk II (Specac, United Kingdom).

For each measurement a 10 μl droplet of sample was placed on the surface of the ATR diamond crystal. 972 samples were measured in total. The OPUS software (Bruker Optik GmbH, Germany) was used for data acquisition and instrument control. Growth media after cultivation were measured with the HTS-XT system as well, in the configuration mentioned above.

**1.4.3. Transmission electron microscopy (TEM) of *Mucor circinelloides* hyphae sections.** Fresh washed fungal biomass was fixated by applying the fixating solution consisting of 2% paraformaldehyde, 1.25% glutaraldehyde and 0.1 M sodium cacodylate buffer for 1 hour at 4˚C. Subsequently, the fixating solution was removed by centrifugation at 11000 rpm for 15 min and the fixated biomass was washed three times with 0.1 M sodium cacodylate buffer (for 10 min at 4˚C for each washing step). Buffer was removed and the fixated biomass was postfixated in 1% OsO4 in 0.1 M sodium cacodylate buffer for 1h. After the postfixation, the fungal biomass was dehydrated with ethanol employing each of the following ethanol concentrations for 15 minutes: 70%, 90%, 96% and 100% ethanol. The last concentration was repeated four times for 15 minutes. Thereafter, the LR White resin (LRW) medium grade was infiltrated into the biomass in the mixture with ethanol in following LRW/ethanol ratios: 1:3, over night; 1:1 overnight; 3:1 overnight; 100% LRW overnight. Finally, the fixated biomass was embedded in 100% LRW overnight at 60˚C in the oven. The embedded biomass was sectioned using Leica EM UC6 into 60 nm thin slices and sections were monitored using FEI Morgagni 268 Transmission electron microscope equipped with Olympus Veleta CCD camera.

## 1.5. Data analysis

The Following software packages were used for the data analysis: Unscrambler X version 10.5.1 (CAMO Analytics, Norway), Orange data mining toolbox version 3.15 (University of Ljubljana, Slovenia) [78, 79], and Matlab R2018a (The Mathworks Inc., Natick, USA).

**1.5.1. Pre-processing of FTIR spectra.** The pre-processing of FTIR-HTS spectra was performed in two ways:

(1) FTIR-HTS spectra of the biomass were first transformed to second-derivative spectra by the Savitzky–Golay algorithm using a polynomial of degree 2 and a window size of either 11 or 61 points in total. Different window sizes were used in order to emphasize either narrow peaks associated with lipids and chitin/chitosan (window size 11), or broad peaks associated with polyphosphates (window size 61). The second-derivative spectra were pre-processed by extended multiplicative scatter correction (EMSC), an MSC model extended by a linear and quadratic components [80–82]. Technical replicates (936 spectra in total) were averaged in order to remove technical variability of the measurements, resulting into 312 spectra. These spectra were cut and used for the PCA analysis of specific lipid- (3020–2819 $cm^{-1}$, 1760–1726 $cm^{-1}$, 1475–1375 $cm^{-1}$, 1160–1149 $cm^{-1}$, 730–715 $cm^{-1}$), polyphosphates- (1301–1203 $cm^{-1}$, 925–842 $cm^{-1}$) and chitin/chitosan (3457–3417 $cm^{-1}$, 3293–3251 $cm^{-1}$, 3133–3081 $cm^{-1}$, 1639–1623 $cm^{-1}$, 1392–1346 $cm^{-1}$, 962–941 $cm^{-1}$) spectral regions (Figs 7, 9 and 11) in order to show the reproducibility of the growth experiment (i.e. biological replicates).

(2) In order to get overview of all samples in whole measured spectral region, technical replicates (936 spectra in total) were averaged in order to remove technical variability of the measurements, resulting into 312 spectra (biological replicates). Further, biological replicates were averaged, resulting in 104 FTIR spectra, and pre-processed by EMSC. After pre-processing, spectra were used for PCA analysis (Fig 6) and to plot each fungal strain separately for the observation of the effect of different amounts of phosphate salts on the biochemical composition of biomass (S1–S18 Figs).

**1.5.2. Principle component analysis (PCA) and variation contribution analysis.** Principle component analysis (PCA) was conducted on the pre-processed FTIR data. To evaluate

influence of different nitrogen sources, PCA analysis was done on the data set split into two parts: 1) samples grown on yeast extract (YE), 2) samples grown on ammonium sulphate (AS). Variation in the data introduced by the different design parameters, specifically N-source, Pi concentration and N-Pi interaction, was calculated for each strain independently in each data set. In ANOVA model a data matrix is represented as a sum of matrices that describe experimental design factors and the residual error. Each of these matrices consists of the means of the spectra that correspond to different levels of the design factor. The variation due to each factor can then be calculated. The ANOVA model for this study contained three design factors: N-source, Pi concentration and N-Pi interaction. The factor "N-source" had two levels (YE, AS), the factor "Pi concentration" consisted of six levels (six different Pi concentrations), the design factor "N-Pi interaction" had 12 levels. Biological and other variations not of interest for this study were kept as a part of residuals. The variation of each factor was normalized by the sum of the variations for the three factors of interest, so they summed up to 100%. Such ANOVA model underlies commonly used ANOVA-PCA and ASCA analysis [83, 84] which in addition to calculating variation contribution of design factors in a data allow analyzing other aspects of the data. The methods were therefore not implemented in this study.

**1.5.3. Monitoring of glucose and phosphate consumption.** FTIR-ATR spectra of growth media after the cultivation were used for the estimation of glucose and phosphate consumption. ATR spectra of pure water, and water solutions of pure glucose, nitrogen sources, phosphates, as well as media before cultivation were evaluated for characteristic signal of the components (S19 Fig). The peak at 1799 cm$^{-1}$ was selected for the correction of baseline shift, while the peak associated with water at 1637 cm$^{-1}$ was selected for peak normalization of all spectra. All growth media spectra were first baseline corrected ($A_{nv}$-$A_{n1799}$), and then peak normalized ($A_{nv}$/ $A_{n1637}$), where $A_{nv}$ is the absorbance value of sample *n* at a specific wavenumber. Finally, growth media spectra were corrected for water absorbance by subtracting the absorbance values of baseline- corrected and peak- normalized water spectrum from the corresponding absorbance values of the preprocessed growth media spectra. The peak associated with glucose at 1034 cm$^{-1}$ ($A_{n1034}$) and peak associated with phosphates at 937 cm$^{-1}$ ($A_{n937}$) were used to estimate nutrient consumption in the growth media. Phosphate consumption ($P_{FTIR}$) was based on the $A_{n937}$ value of the growth media. Glucose consumptions ($G_{FTIR}$) was calculated according to the equation:

$$\mathbf{G_{FTIR} = A_{GM1034} - A_{GM937}\frac{A_{P1034}}{A_{P937}}}$$

where $A_{GM}$ and $A_{P}$ are the absorbance values (at the corresponding wavenumbers) for pre-processed growth media spectrum and water solution of pure phosphate spectrum, respectively. The second term in the equation is taking into consideration that both glucose and phosphate contribute to the absorbance at 1034 cm$^{-1}$ (i.e. the term is estimating phosphate contribution to the total absorbance at 1034 cm$^{-1}$ based on measurement of water solution of pure phosphate). Four media samples, belonging to one biological replicate of *Umbelopsis vinacea* grown in ammonium sulphate with Pi8, Pi4, Pi2, and Pi1 phosphate concentrations, were excluded from the analysis due to a technical error in the preparation of the samples for the FTIR measurements.

# 2. Results

## 2.1. Growth characteristics of *Mucoromycota* fungi under different nutrient conditions

**2.1.1. Biomass production and pH.** Two types of nitrogen (N) sources, yeast extract (YE) and ammonium sulphate (AS), and six concentrations of inorganic phosphorus (Pi) were

applied to study the effect of N source and Pi level on the nutrient-induced co-production of high-value metabolites–lipids, chitin/chitosan and polyphosphates, in *Mucoromycota* fungi.

Fig 1A shows the effect of yeast extract- complex organic multi-component substrate containing both nitrogen and phosphorus, on the cultivation of *Mucoromycota* fungi under different Pi levels. Results indicate that the addition of inorganic phosphorus could be neglected, since it does not have any significant effect on the biomass production. Yeast extract contains approximately 2.5% of total phosphorus. This amount corresponds to approx. 15% in terms of total P contained in added phosphates salts in the lowest examined Pi condition- Pi0.25. The highest biomass yield (18.92–23.67 g/L) was observed for *Umbelopsis vinacea* on both types of nitrogen sources (Fig 1). In case of YE-Pi medium, high *Umbelopsis vinacea* biomass yield was obtained for a wide range of phosphorus concentrations (Pi0.5 –Pi4), while on AS-Pi media Pi2 and Pi4 concentrations showed the highest biomass yield. This indicates that *Umbelopsis vinacea* requires quite high concentration of phosphorus for optimal growth in ammonium sulphate media under nitrogen-limited conditions. The lowest biomass yield was obtained for *Mortierella alpina* on both YE–(5.55–6.10 g/L) and AS-based (0.90–7.03 g/L) media. The biomass yield for *Mucor circinelloides*, *Absidia glauca*, *Lichthemia corymbifera*, and *Amylomyces rouxii* was in a range from 8.52 to 12.92 g/L when grown on yeast extract, and from 2.41 to 12.34 g/L when grown on ammonium sulphate. *Cunninghamella blakesleeana*, *Rhizopus stolonifer* and *Mortierella hyalina* had biomass yields from 4.28 to 9.20 g/L when grown on YE-Pi media, and from 2.11 to 10.05 g/L when grown on AS-Pi media (Fig 1).

The use of different phosphorus (Pi) concentrations resulted in a change of the pH in the media after the cultivation for all studied fungi when ammonium sulphate was used as a nitrogen source. Low phosphorus concentrations caused a significant drop of pH in media for all fungi (Fig 2). In media with yeast extract as a nitrogen source, quite stable pH values where observed for media of several fungi throughout Pi concentration range: *Lichtheimia corymbifera*, *Umbelopsis vinacea*, *Mortierella alpina* and *Mortierella hyalina*. Significantly lower pH values were detected for low Pi concentrations compared to the media with higher Pi concentrations for *Mucor circinelloides*, *Absidia glauca*, *Cunninghamella blakesleeana*, *Amylomyces rouxii* and *Rhizopus stolonifer*. Thus, YE shows higher buffering capacity than AS, which was confirmed by titration of YE-Pi0.25 and AS-Pi0.25 growth media with 1M HCl (S20 Fig).

## 2.2. Fourier Transform Infrared (FTIR) spectroscopy reveals co-production in oleaginous *Mucoromycota* fungi

FTIR spectroscopy is a non-destructive technique that allows examining the total biochemical profile of intracellular metabolites in fungal cells, as well as extracellular metabolites, by using high-throughput screening (HTS) FTIR measurements. Moreover, monitoring of growth media components (glucose and phosphates) was obtained by using attenuated total reflectance (ATR) FTIR measurements. In infrared spectroscopy, the loss of infrared radiation due to chemical absorption is quantified. In the FTIR-HTS transmission mode, the loss of radiation due to absorption is quantified by transmitting infrared radiation through a sample and quantifying the loss of the radiation by comparing the transmitted radiation with the radiation that impinges on the sample. By covering the complete spectra range of the mid-infrared, biochemical fingerprint of all major chemical building blocks is obtained. The FTIR-HTS system employs a high-throughput setup with microplates and automated measurements allowing the automated analysis of around 180 samples in one measurement run. Relatively large variance in sample thickness results in the difference in optical path length, which can be corrected by standard pre-processing tools developed by us [80, 82]. In FTIR-ATR analysis, the infrared radiation undergoes reflection in an ATR crystal an produces an evanescent field in the sample

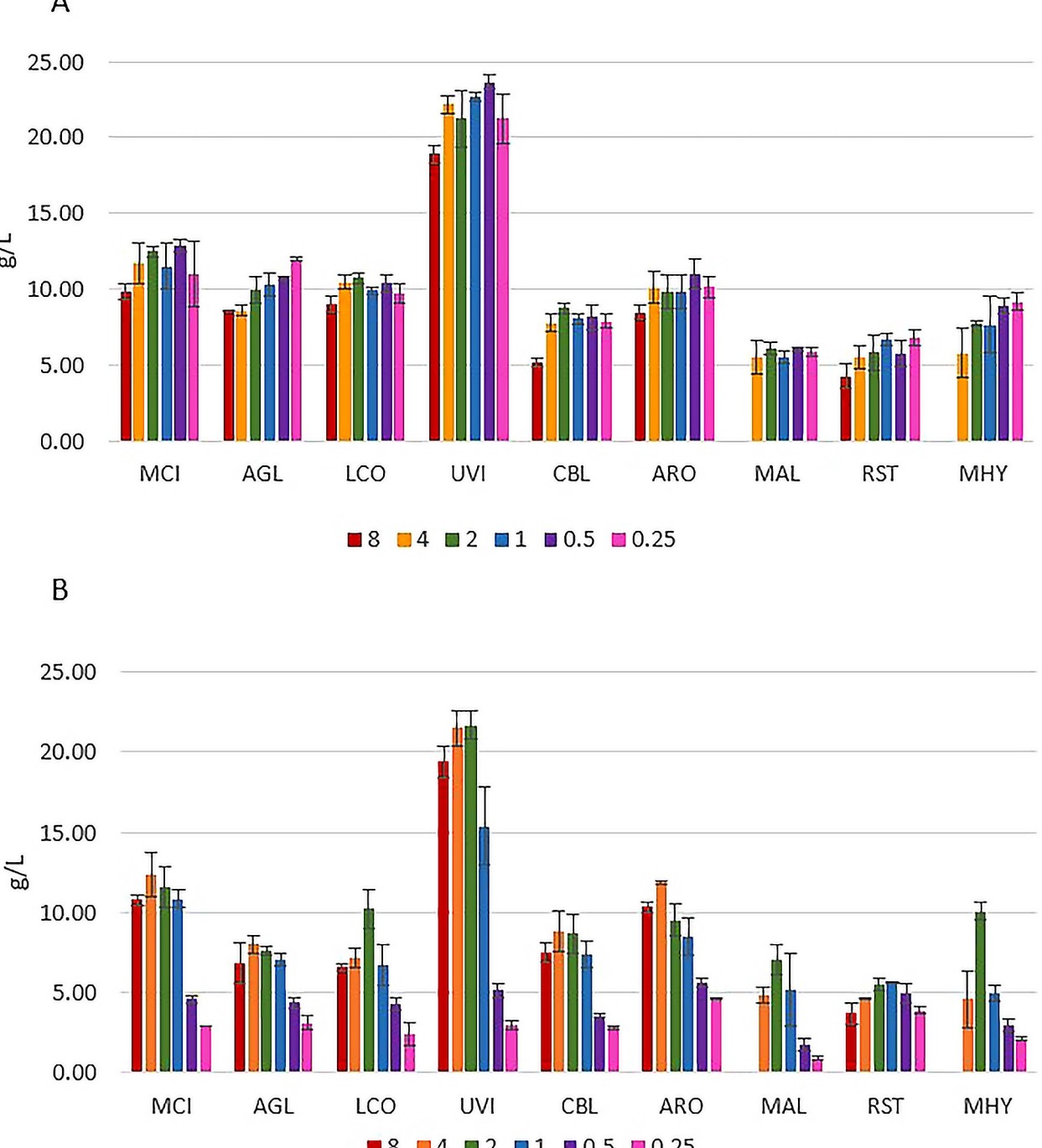

**Fig 1.** Biomass yield for *Mucoromycota* fungi grown in media with (A) yeast extract (YE) and (B) ammonium sulphate (AS) under different Pi concentrations. Different colors correspond to different Pi concentrations (Table 2). *Absidia glauca-* AGL, *Amylomyces rouxii-* ARO, *Cunninghamella blakesleeana-* CBL, *Lichtheimia corymbifera-* LCO, *Mortierella alpina-* MAL, *Mortierella hyalina-* MHY, *Mucor circinelloides-* MCI, *Rhizopus stolonifer-* RST, *Umbelopsis vinacea-* UVI.

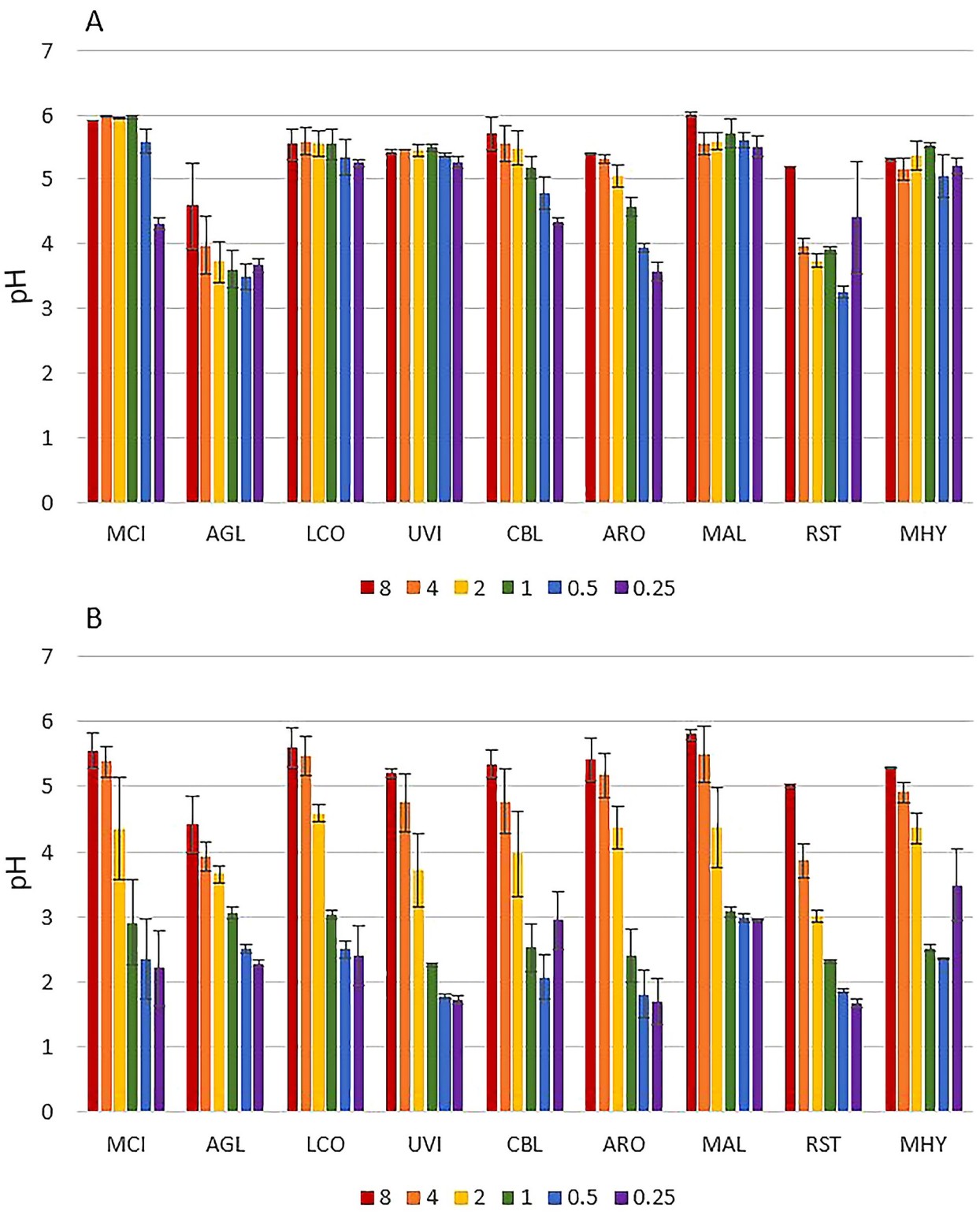

**Fig 2.** pH values of YE-Pi media with different phosphorus (Pi) levels after the cultivation on YE-Pi (A) and AS-Pi (B) media. The initial pH value at the start of the cultivation was 6±0.3. Different colors correspond to different Pi concentrations (Table 2). *Absidia glauca*- AGL, *Amylomyces rouxii*- ARO, *Cunninghamella blakesleeana*- CBL, *Lichtheimia corymbifera*- LCO, *Mortierella alpina*- MAL, *Mortierella hyalina*- MHY, *Mucor circinelloides*- MCI, *Rhizopus stolonifer*- RST, *Umbelopsis vinacea*- UVI.

which is located on its surface. The evanescent field is attenuated by the sample die to chemical absorption and the absorption can be quantified by relating the attenuated radiation with the radiation that is obtained in an ATR setup without a sample at the surface of the crystal. The ATR setup is characterized by a high reproducibility caused by a stable penetration depth of the IR beam into the sample, when the sample at the top of the crystal is in tight contact with the surface of the crystal. This is true for liquid and viscous samples such as the culture supernatant in our measurements. Information about intracellular and extracellular metabolites of fungal cells could be read from different spectral regions of HTS FTIR spectra (Fig 3, Table 3): (1) The region from 3010–2800 $cm^{-1}$, 1800–1700 $cm^{-1}$ and some single peaks related to $-CH_2$ and $-CH_3$ scissoring in a region ~1460 $cm^{-1}$ contain detailed information about lipids. One of the most important lipids associated peaks is ~1745 $cm^{-1}$ which is related to the carbonyl bond stretching in esters and indicating the lipid (acylglycerols) content in the cell. The peak around 1715 $cm^{-1}$ is related to the carbonyl bond vibrations in organic acids, and the peak around 3010 $cm^{-1}$ is related to = C-H stretching in lipids and indicating the unsaturation level of lipids in the cell. (2) Proteins have peaks in the region from 1700–1500 $cm^{-1}$; (3) Polyphosphates– 1260–1240 $cm^{-1}$ and 885–880 $cm^{-1}$; (4) Glucans peaks can be found in the region 1160–1050 $cm^{-1}$; (5) Chitin/chitosan shows peaks in the region 3440–3100 $cm^{-1}$ and a single peak at 1377 $cm^{-1}$. A more detailed overview of characteristic peaks can be found in Table 3. For the ATR FTIR monitoring of media (Fig 4), the most important peaks were related to glucose (1151, 1103, 1080, 1034 and 990 $cm^{-1}$) and phosphates (1161, 1076 and 937 $cm^{-1}$).

Fourier transform infrared (FTIR) spectroscopy can provide both qualitative and quantitative measures. Quantitative analysis by FTIR requires regression onto reference data. For regression analysis often methods based on latent variables such as partial least square regression are used. As reference data for respective metabolites, e.g. chromatography analyses can be used. Qualitative measures are achieved by spectral assignments (see Fig 3 and Table 3) and by applying unsupervised multivariate data analysis tools (for example principal component analysis or ANOVA-PCA). Although FTIR spectroscopy cannot provide absolute quantifications without establishing calibration models based on reference quantitative data, a semi-quantitative analysis of ratios of chemical constituents (see Fig 13) can be obtained. Nevertheless, the biggest advantage of the FTIR approach is that it allows high-throughput screening of samples and detection of a vast range of different metabolites simultaneously within a single analytical run. Thus, it provides high precision qualitative information allowing to pre-select strains and growth conditions.

The FTIR spectra of *Mucor circinelloides*, grown on a AS nitrogen-source, illustrate the effect of phosphorus availability in media on the intracellular production of lipids, polyphosphates and chitin/chitosan (Fig 3A). Signals of these metabolites clearly correspond to the model components- chitin, glyceryl trioleate and sodium polyphosphate (Fig 3B). *Mucor circinelloides* showed good oleaginous properties when phosphorus was not limited (Pi1 –Pi8), as indicated by strong absorbance peaks related to acylglycerides (3010, 2925, 2854, 1743, and 725 $cm^{-1}$). Moreover, an increase in the amount of phosphorus in the growth media (Pi2 –Pi8) led to increased polyphosphates accumulation in fungal cells, as indicated by the strong absorbance peaks related to polyphosphates (1265 and 883 $cm^{-1}$). FTIR results clearly indicated that limitation of phosphorus availability (Pi0.25 and Pi0.5) resulted in low pH and an overproduction of chitin/chitosan which could be explained as an activation of protective mechanisms in the cell wall. The production of chitin and chitosan is strongly supported by an observation of the absorbance peaks related to these biopolymers at 3434, 3274, 3104, 1660, 1629, 1550, 1377, and 952 $cm^{-1}$. FTIR-HTS spectra of all strains used in the study can be found in the supplementary materials.

The FTIR-HTS spectra of media after growth, in particular of the AS nitrogen-source media, show carbonyl peaks (at approx. 1715 $cm^{-1}$) (Fig 4A). These carbonyl peaks may relate

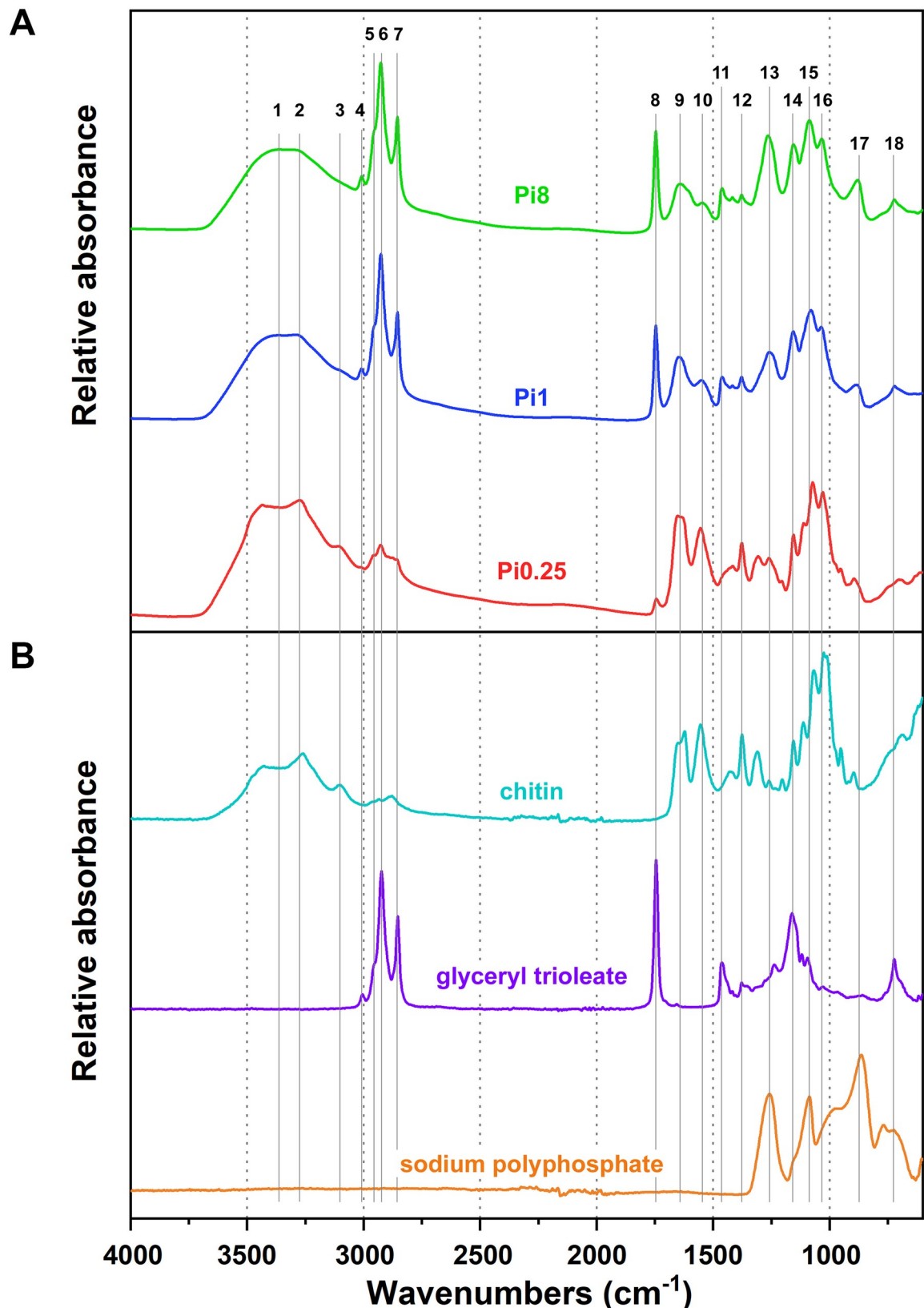

**Fig 3. FTIR-HTS spectra of fungal biomass and model compounds.** A) Preprocessed FTIR-HTS spectra of *Mucor circinelloides* biomass grown on ammonium sulphate and different Pi levels. B) Preprocessed FTIR-ATR spectra of model compounds: chitin, glyceryl trioleate and sodium polyphosphate. Spectra are plotted with an offset for better viewing. Peak numbers correspond to the numbers given in Table 3.

**Table 3. Peak assignments of the FTIR spectra of the fungal cells (chemical class with the predominant contribution is stated in the parenthesis).**

| Peak Nr. | Wavenumber (cm⁻¹) | Peak assignment | Reference |
|---|---|---|---|
| 1 | 3500–3200 | O-H stretching (carbohydrates) | [85] |
| 2 | 3275 | N-H stretching (chitin/chitosan) | [85] |
| 3 | 3105 | N-H stretching (chitin/chitosan) | [85] |
| 4 | 3010 | = C-H stretching (lipids) | [86] |
| 5 | 2955 | -C-H ($CH_3$) stretching (lipids) | [86] |
| 6 | 2925 | >$CH_2$ of acyl chain (lipids) | [86] |
| 7 | 2855 | -C-H ($CH_2$) stretching (lipids) | [86] |
| 8 | 1745 | -C = O stretching in esters (lipids) | [86] |
| 9 | 1680–1630 | -C = O stretching, Amide I (proteins, chitin) | [87, 88] |
| 10 | 1530–1560 | C-N-H deformation, Amide II (proteins, chitin) | [88, 89] |
| 11 | 1465 | -C-H ($CH_2$, $CH_3$) bending (lipids) | [86] |
| 12 | 1377 | -C-H ($CH_3$) bending (chitin) | [86] |
| 13 | 1265 | P = O stretching (polyphosphates) | [66] |
| 14 | 1160 | C-O-C stretching in esters (lipids) | [90] |
| 14–16 | 1200–1000 | C-O and C-O-C stretching (carbohydrates) | [91] |
| 17 | 885 | P-O-P stretching (polyphosphates) | [66] |
| 18 | 725 | >$CH_2$ rocking in methylene–$(CH_2)_n$-chains (lipids) | [86] |

to production of organic acids coming from the Krebs cycle, for example citric acid. This is in agreement with pH measurements (Fig 2B) and with our previous studies [54], where citric acid was determined by HPLC measurements. In order to confirm this observation, transmission electron microscopy (TEM) of the *Mucor circinelloides* sectioned hyphae, obtained from the growth on ammonium sulphate media with Pi-limited and Pi-non-limited conditions, was performed (Fig 5).

As it can be seen from TEM images, the cell wall of *Mucor circinelloides* hyphae grown on Pi-limited condition (Fig 5A) is much thicker than the cell wall of the hyphae from Pi-non-limited condition (Fig 5B), while size and number of lipid bodies are smaller in a Pi-limited than in Pi-non-limited conditions. This indicates the increase in the cell wall components–chitin/chitosan and decrease in the lipid accumulation for the hyphae obtained from Pi-limited conditions is in accordance with the FTIR-HTS spectroscopy results reported above.

**2.2.1. The influence of N-source and Pi-levels on the co-production in *Mucoromycota* fungi.** A nitrogen (N) source used for the fungal fermentation can be organic or inorganic. In this study, yeast extract (YE) was used as an organic N-source and ammonium sulphate (AS) as an inorganic N-source. PCA analysis of FTIR-HTS spectral data was performed to reveal the biochemical composition of the samples.

The PCA score plot of the first and second component of FTIR-HTS spectra of fungal biomass grown on YE is shown in Fig 6A, the corresponding loadings are shown in Fig 6C. The PCA score plot shows clear strain-specific clustering. Higher components did not show relevant trends related to the main biomass constituents. This indicates that each fungus has its strain-specific biochemical composition when grown on YE. Different Pi concentrations are not influencing these strain-specific fingerprints considerably. The loadings in Fig 6C show that the strain-specific differences in biomass composition are mostly determined by the ratio of main cellular components, specifically lipids (3010, 2925, 2855 cm⁻¹), polyphosphates (1265, 885 cm⁻¹), chitin/chitosan (3434, 3275, 3105, 1660, 1629, 1550, 1377, and 952 cm⁻¹) and proteins (1680–1630, 1530–1560 cm⁻¹). For example, biomass of *Mucor circinelloides* and *Amylomyces rouxii* have high phosphate (polyphosphates) to nitrogen (chitin, chitosan and proteins)

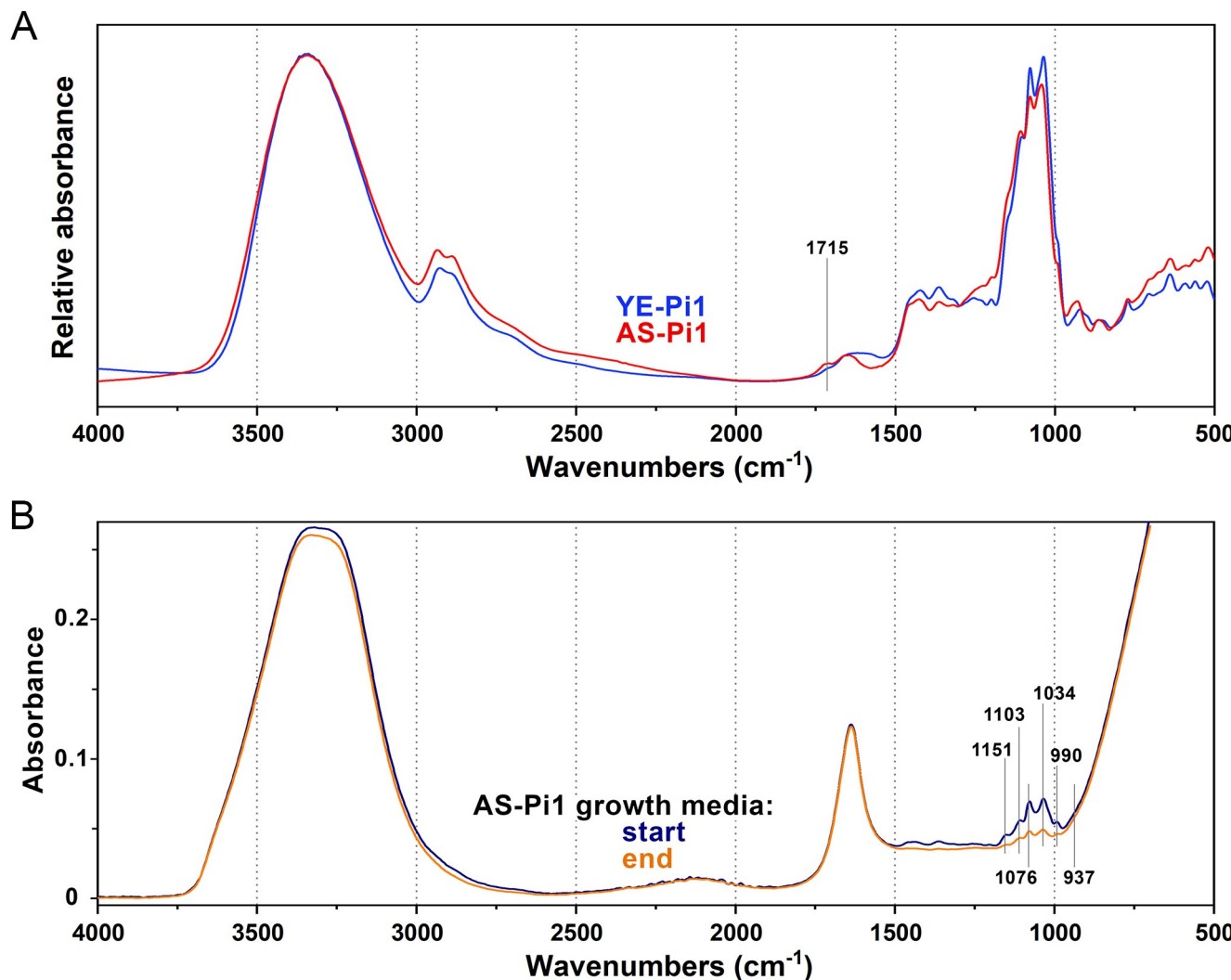

**Fig 4. FTIR spectra of growth media.** A) Preprocessed FTIR-HTS spectra of ammonium sulphate and yeast extract growth media (both Pi1) at the end of cultivation of *Mucor circinelloides*, B) FTIR-ATR spectra of ammonium sulphate Pi1 growth media at the beginning and end of cultivation of *Mucor circinelloides*.

ratio, compared to *Cunninghamella blakesleeana* and *Lichtheimia corymbifera*. Compared to all of them, spectra of *Umbelopsis vinacea* show the highest lipids-to-proteins ratio.

The PCA score plot of the first and second component of FTIR-HTS spectra of fungal biomass grown on AS is shown in Fig 6B and the corresponding loadings are shown in Fig 6D. In contrast to the FTIR spectra of fungi grown on YE, the FTIR spectra of fungi grown on AS do not show any clustering with respect to fungal strain (Fig 6B). However, unlike for the YE-Pi media, strong biochemical differences for fungi grown in AS-Pi at different phosphorus levels can be clearly seen. The low effect of phosphorus on the biochemical composition of the strains when grown in yeast extract may be explained by the fact that yeast extract is a complex and rich source of not only nitrogen, sulphur, vitamins and minerals, but also of organic phosphorus. Due to a relatively large starting amount of organic phosphorus in the yeast extract, variation in the concentration of the inorganic phosphorus may not have strong effects on fungal growth in the YE media. In the case of AS-based media, when Pi was the only source of phosphorus for fungal growth, considerable changes in fungal cell chemistry were observed when

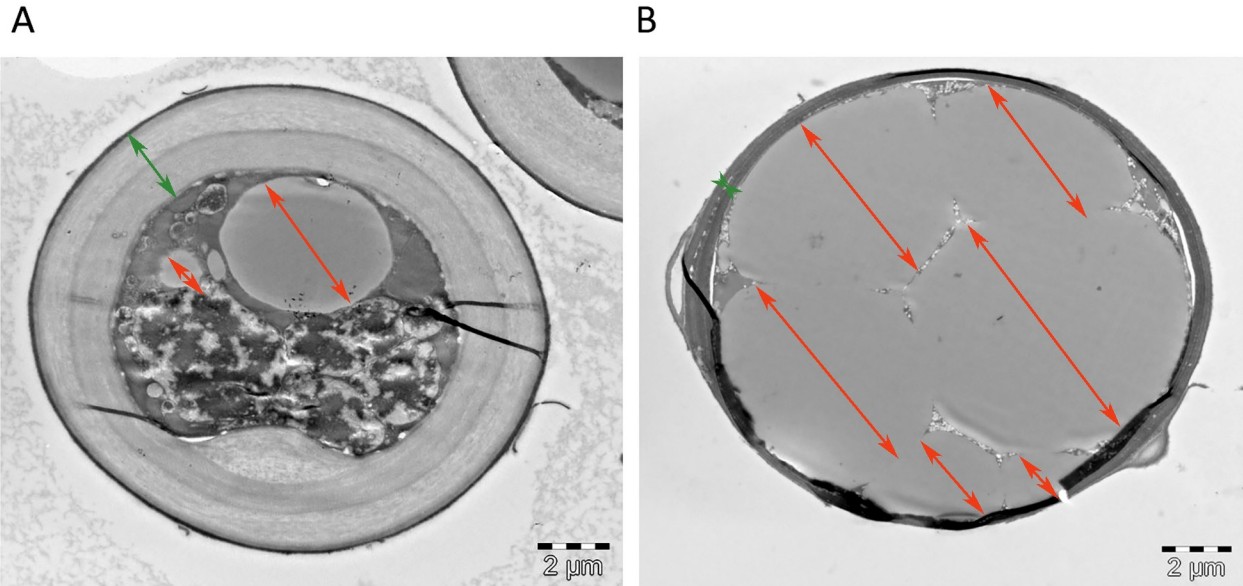

**Fig 5.** Transmission electron microscopy (TEM) of cross sectioned *Mucor circinelloides* (MCI) hyphae grown on AS media with Pi0.5 (limited) (A) and AS Pi1 (non-limited) (B) conditions. Green arrows indicate cell wall and orange arrows lipid bodies. Images are taken by Lene Cecilie Hermansen, Imaging center NMBU.

phosphorus levels were changed. Therefore, biochemical differences in the biomass were quite pronounced at already low Pi concentrations (Fig 6A and 6B).

PCA of FTIR-HTS spectra using specific spectral regions that are characteristic for lipids, polyphosphates and chitin/chitosan was performed in order to evaluate the co-production of these components in *Mucoromycota* fungi (Figs 7, 9 and 11). In order to have a deeper understanding on the influence of variations in N-source, Pi concentration and N-Pi interaction on the co-production of lipids, polyP and chitin/chitosan in *Mucoromycota* fungi, Analysis of Variance PCA (ANOVA-PCA) was performed following the approach by Harrington [83]. The analysis of variation in the FTIR-HTS spectra introduced by the different design factors was done using respective spectral regions (Figs 8, 10 and 12).

In Fig 7A, the score values of the first principal component of the lipid region are shown for all strains and the corresponding loading vector is shown in Fig 7C. From the spread of the score values of the first PC, we can see that availability of inorganic phosphorus did not influence the accumulation of lipids in *Umbelopsis vinacea*, *Mortierella hyalina*, *Mucor circineloides*, and only some minor effects could be seen for *Mortierella alpina* (Fig 7A). Thus, results indicate that addition of inorganic phosphorus might be not needed, when complex organic multi-component substrates containing both nitrogen and phosphorus are used for the production of lipids by *Mucoromycota* fungi. In this case, the addition of Pi does not have any significant effect on the biomass and lipid yield. Moreover, in some cases high levels of phosphorus can negatively influence accumulation of lipids, as it was observed for *Cunninghamella blakesleeana*, *Amylomyces rouxii* and *Absidia glauca* (Fig 7A) which is explained by the growth inhibition effect of high Pi-levels of these fungal strains and the accumulation of polyphosphates in case of *Amylomyces rouxii* and *Absidia glauca*. The observed variation in lipid content of *Rhizopus stolonifer* which is not correlated with Pi availability can be explained by a low relative amount of lipids in *Rhizopus stolonifer* biomass, as shown in Fig 6A.

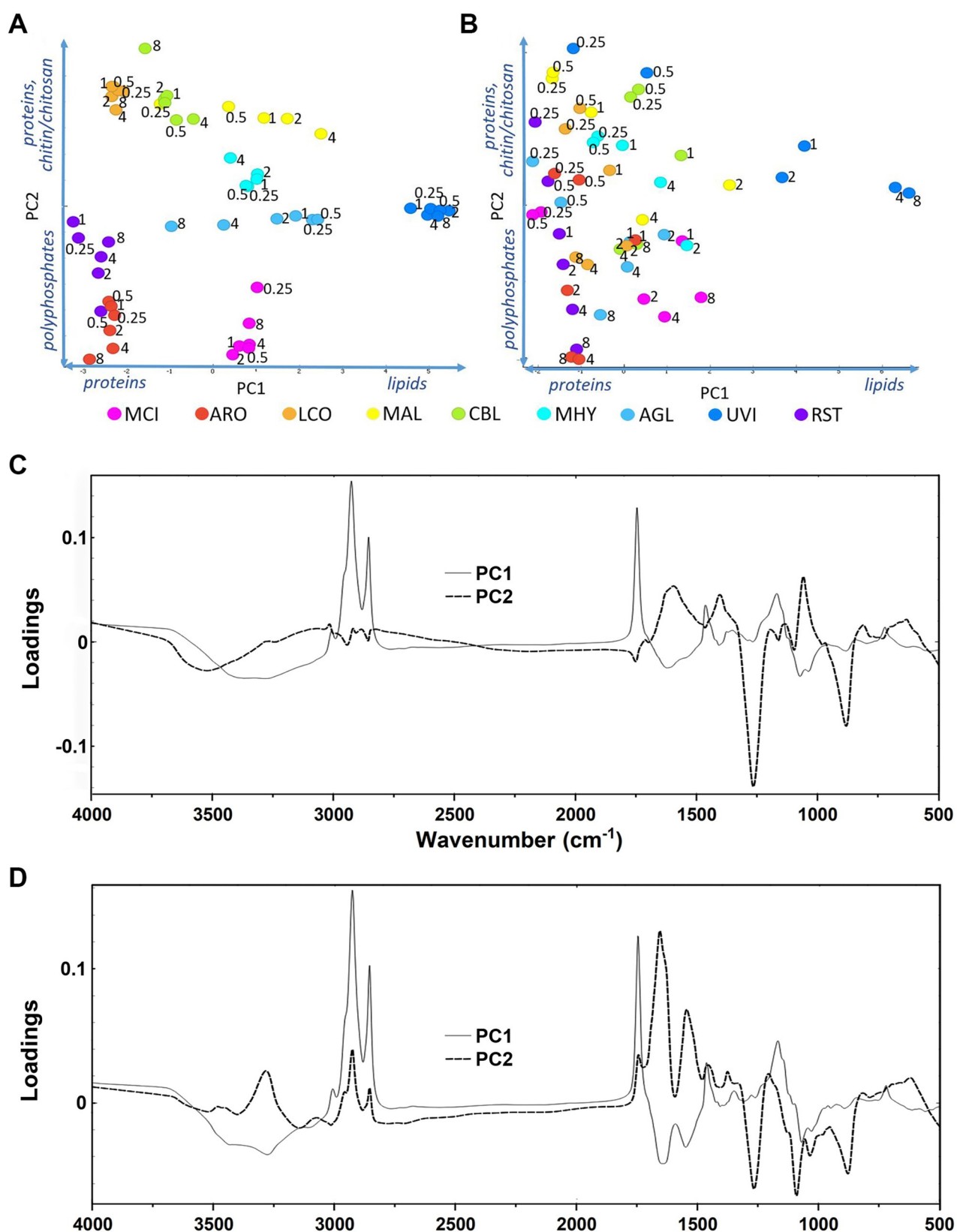

**Fig 6.** PCA score plots of FTIR-HTS spectra of fungi grown at different phosphorus concentrations on yeast extract (A) and ammonium sulphate (B). Numbers in PCA score plots indicate the Pi amounts. Vectors on axis describe an observed increase of the metabolites chitin/chitosan, polyphosphates and lipids. Below the scatter plots, loading vectors for PC1 (full line) and PC2 (dashed line) are plotted in C and D, respectively. The explained variance for the first and second principal components are 87% and 7%, respectively, for YE and 69% and 20% for AS.

The score values of the first principal component of the PCA of the lipid region of FTIR spectra of AS-Pi grown fungi is shown in Fig 7B and the corresponding loading vector in Fig 7D. The spread of the score values indicates that lipid accumulation in *Absidia glauca*, *Amylomyces rouxii* and *Mucor circinelloides* was stronger influenced by the Pi level than in other fungi. The decrease in Pi in AS-Pi media led to the low lipid content for all fungi except *Cunninghamella blakesleeana* (Fig 7B). *Mucor circinelloides* showed the highest decrease in the

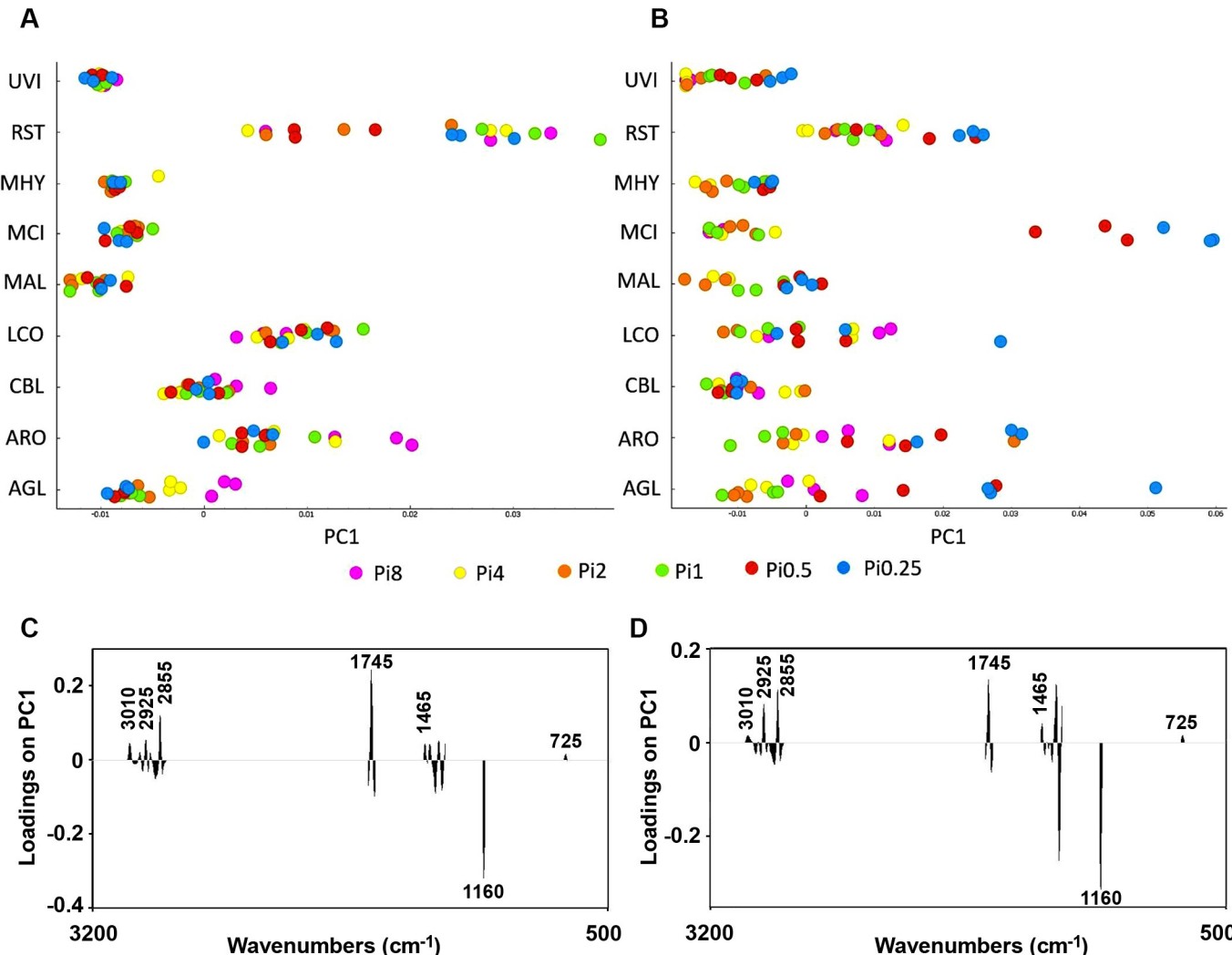

**Fig 7. PCA results (first principal component) for the lipid region 3020–2819 cm⁻¹, 1760–1726 cm⁻¹, 1475–1375 cm⁻¹, 1160–1149 cm⁻¹, 730–715 cm⁻¹) of FTIR-HTS spectra (pre-processed by 2ⁿᵈ derivative and EMSC).** The scores for the first principal component are plotted for all strains in A and B. In A and C, the score plot and the corresponding loading plot are shown for fungi grown on YE-Pi using different Pi levels. In B and D, the score plot and corresponding loading plot are shown for fungi grown on AS-Pi media using different Pi levels. The color coding is according to the Pi levels. The loading plots show that the total lipid content is increasing from the right to the left in both score plots. The explained variance for the first principal component is 66% and 76% for YE and AS, respectively.

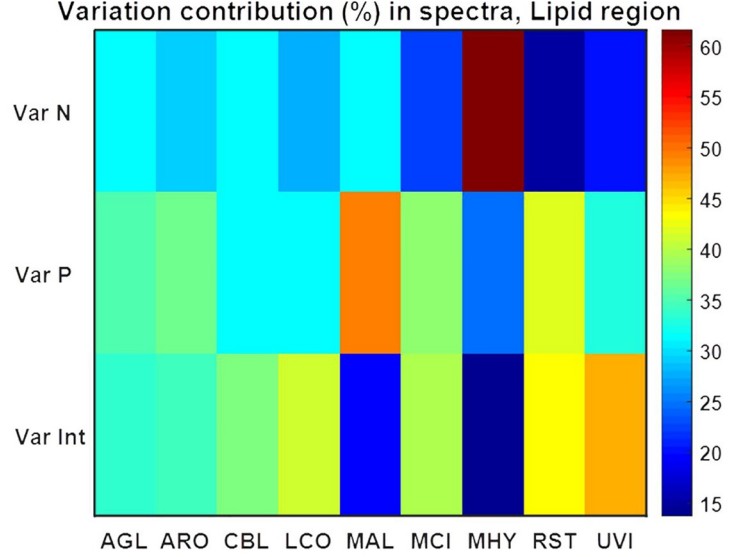

**Fig 8. Variation contribution (%) from changes in N, Pi and N-Pi interaction on the lipid region (3020–2819 cm$^{-1}$, 1760–1726 cm$^{-1}$, 1475–1375 cm$^{-1}$, 1160–1149 cm$^{-1}$, 730–715 cm$^{-1}$) of FTIR-HTS spectra.** Variation contributions due to changes in N and Pi alone are presented in the first two rows (Var N and Var P), whereas contribution from N-Pi interaction (Int) is presented in the last row (Var Int).

lipid accumulation at low Pi levels (Pi0.5 and Pi0.25). The reason for this is that both cell growth as well as the lipid accumulation process involves a set of phosphorylated molecules, the synthesis of which could be inhibited under Pi-limited conditions. Generally, Pi concentrations Pi1, Pi2 and Pi4 where better suited for the lipid accumulation in *Mucoromycota* fungi (Fig 7B). Based on the biomass yield results, the optimal phosphorus amount for the fungal growth in growth media that are poorer in nutrients is Pi2 for *Mortierella*. Taking in consideration that *Mortierella* are producing high value polyunsaturated fatty acids, this finding has importance for optimization of industrial bioprocesses. *Lichtheimia corymbifera* did not show any specific trend in lipid content with respect to the amount of Pi neither in YE-Pi, nor AS-Pi media.

ANOVA model for spectral data using the lipid region (Fig 8) (3020–2819 cm$^{-1}$, 1760–1726 cm$^{-1}$, 1475–1375 cm$^{-1}$, 1160–1149 cm$^{-1}$, 730–715 cm$^{-1}$) showed that variation in N, Pi and N-Pi interaction influences fungal lipids in different ways depending on the fungal strain, and the N-source variation had the least influence on the lipid accumulation in all fungi except *Mortierella hyalina* (Fig 8). For *Absidia glauca*, *Amylomyces rouxii* and *Cunninghamella blakesleeana*, variation of nitrogen, phosphorus and their combination influenced the lipid production to the same extent. For *Lichtheimia corymbifera* and *Umbelopsis vinacea* there was a slightly higher influence of the N-Pi interaction than of each of the nutrients separately. The lipid production of *Mucor circinelloides* and *Rhizopus stolonifer* was not strongly affected by the different nitrogen sources, contrary to *Mortierella hyalina*, where the nitrogen source played an important role in lipid accumulation. Variation of phosphorus caused the biggest changes in the lipid production of *Mortierella alpina*.

The co-production of polyphosphate (polyP) and lipids was studied by PCA analysis of the spectral regions of HTS-FTIR spectra of fungi that have characteristic bands from polyphosphate (1301–1203 cm$^{-1}$, 925–842 cm$^{-1}$). The score values of the first principal component of the PCA of the of the spectral regions that have characteristic bands from polyphosphate of fungi grown on YE-Pi media is shown in Fig 9A and the corresponding loading vector in Fig

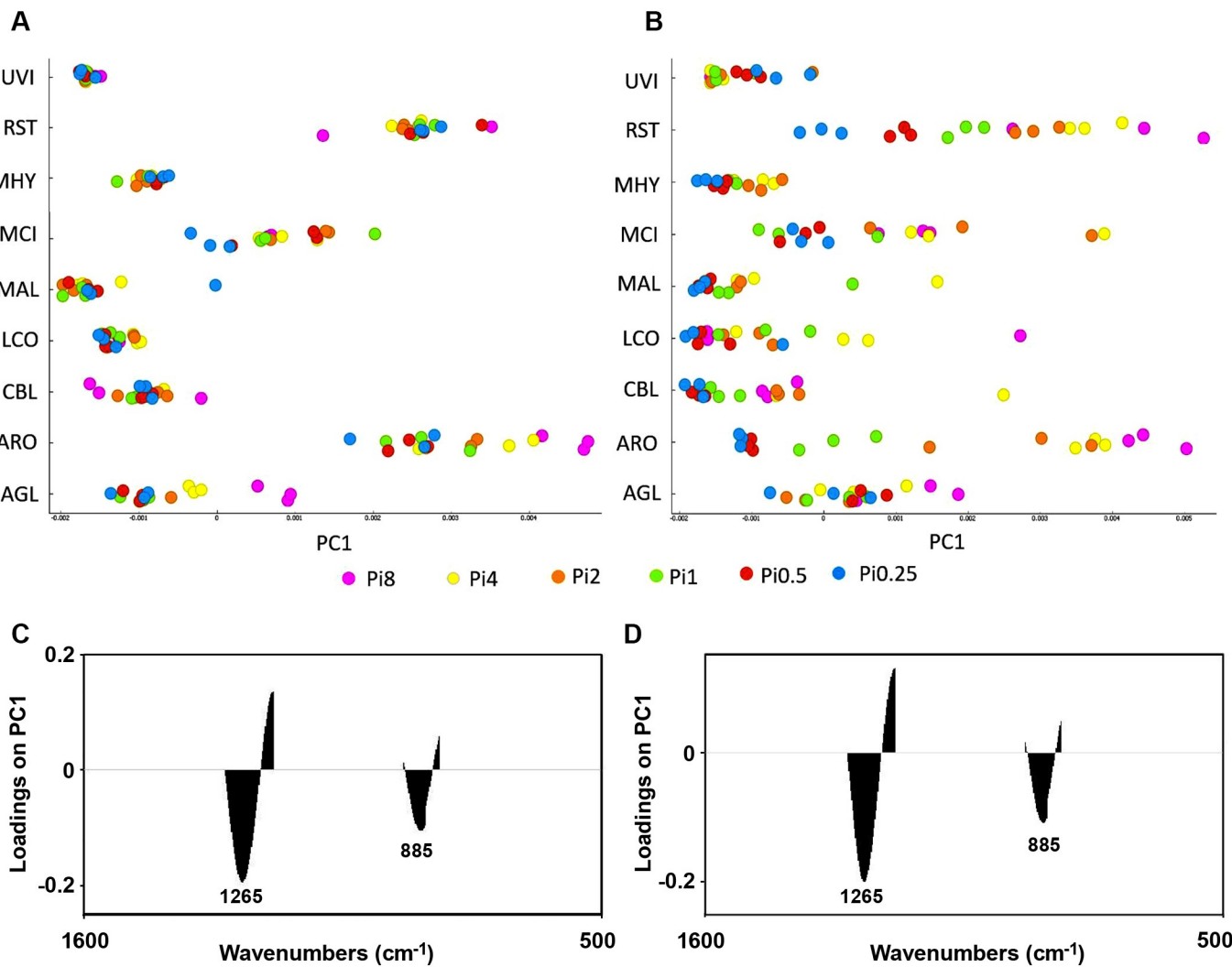

**Fig 9.** PCA results (first principal component) for the polyphosphate region (1301–1203 cm$^{-1}$, 925–842 cm$^{-1}$) of FTIR-HTS (pre-processed by 2$^{nd}$ derivative and EMSC) spectra of fungi grown on (A) YE-Pi and (B) AS-Pi. The scores for the first component is plotted for all strains in A and B. In A and C, the score plot and the corresponding loading plot are shown for fungi grown on YE-Pi using different Pi levels. In B and D, the score plot and corresponding loading plot are shown for fungi grown on AS-Pi media using different Pi levels. The color coding is according to the Pi levels. The loading plots show that the total polyphosphates content is increasing from the left to the right in both score plots. The explained variance for the first principal component is 95% and 93% for YE and AS, respectively.

9C. It was observed that *Rhizopus stolonifer*, *Mucor circinelloides*, *Amylomyces rouxii* and *Absidia glauca* grown in YE-Pi and AS-Pi media show significant polyP accumulation along with lipid accumulation when a high level of Pi was used (Fig 9).

A co-production of polyP in addition to lipids could not be observed for *Mortierella* fungi and *Umbelopsis vinacea*. While, some polyP accumulation was observed for *Umbelopsis vinacea* with low phosphorus in AS-Pi media., probably due to the high salinity. Specifically, relatively high salinity was observed when phosphorus media were depleted, since KCl and NaCl were used to keep the same K/Na ratio in phosphorus limited media as for the standard conditions. Polyphosphates are reported to be involved in the adaptation mechanisms of microorganisms to stress conditions, namely temperature, radiation, or salinity [92, 93].

By analyzing the variance contribution using ANOVA model in the polyP-related spectral region (Fig 10) (1301–1203 cm$^{-1}$, 925–842 cm$^{-1}$) it was possible to identify polyP-accumulating

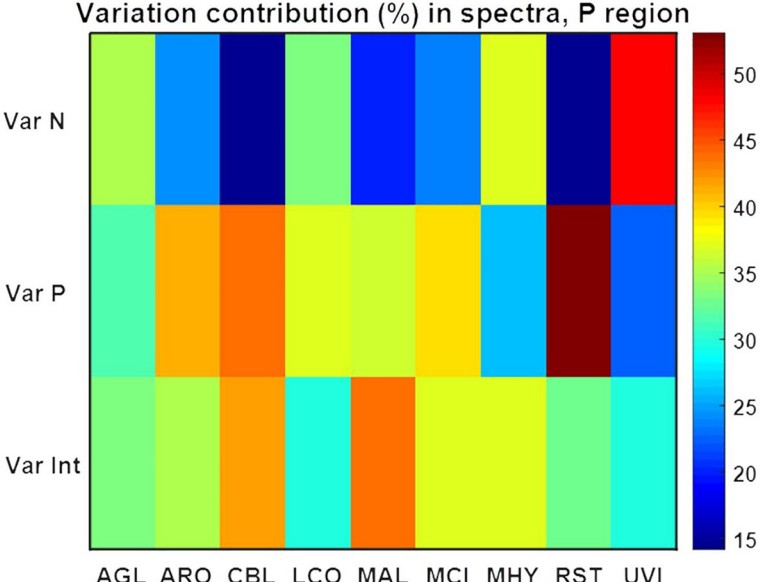

**Fig 10. Variation contribution (%) from changes in N, Pi and N-Pi interaction on the polyphosphate region (1301–1203 cm$^{-1}$, 925–842 cm$^{-1}$) of FTIR HTS spectra.** Variation contributions due to N and Pi alone are presented in the first two rows (Var N and Var P), whereas contribution from N-Pi interaction (Int) is presented in the last row (Var Int).

strains. Increased variation in the polyphosphate region of FTIR-HTS spectra was observed for *Amylomyces rouxii*, *Cunninghamella blakesleana*, *Mucor circinelloides* and most of all *Rhizopus stolonifer*, for which an extraordinary correlation between variation in phosphorus amount in the growth media and intracellular polyphosphates was already noticed in the PCA (Fig 6, Fig 9). For these strains, the phosphorus variation has much more influence on changes in the polyP region of spectra than variation in nitrogen source. The lowest effect of Pi variation was observed for non-polyP accumulating fungi *Mortierella hyalina* and *Umbelopsis vinacea* (Fig 10). The influence of N-source variation was stronger in the case of spectral region related to polyP than the lipid region for *Absidia glauca*, *Mortierella hyalina*, *Lichtheimia corymbifera* and *Umbelopsis vinacea* (Fig 10). In addition, it could be seen that N-Pi interaction has a higher influence on polyP than on the lipid spectral region. This is associated mainly with Pi variation which occurred in AS-Pi media.

In Fig 11, PCA results (first principal component) are shown for the spectral regions 3457–3417 cm$^{-1}$, 3293–3251 cm$^{-1}$, 3133–3081 cm$^{-1}$, 1639–1623 cm$^{-1}$, 1392–1346 cm$^{-1}$, 962–941 cm$^{-1}$, which show characteristic for chitin/chitosan region of the FTIR-HTS spectra of fungi grown on YE-Pi and AS-Pi. In Fig 11A and Fig 11C the first score and loading are shown for the PCA results for fungi grown on AS-Pi media with Pi concentrations Pi0.5 and Pi0.25. The loading plot (Fig 11C) indicates that chitin and chitosan content increases from the left to the right. In Fig 11B and Fig 11D, the corresponding score plot and loading plot are shown for fungi grown on AS-Pi media. From the score plot in Fig 11B we can see that *Mucor circinelloides*, *Rhizopus stolonifer*, *Amylomyces rouxii*, *Absidia glauca* and *Lichtheimia corymbifera* grown on AS-Pi media with Pi concentrations Pi0.5 and Pi0.25 showed increased content of chitin/chitosan (Fig 11B) while the content of lipids was reduced (Fig 7B). In addition to lipids and polyP, several *Mucoromycota* fungi were able to overproduce chitin/chitosan under phosphorus limited conditions (Fig 11B).

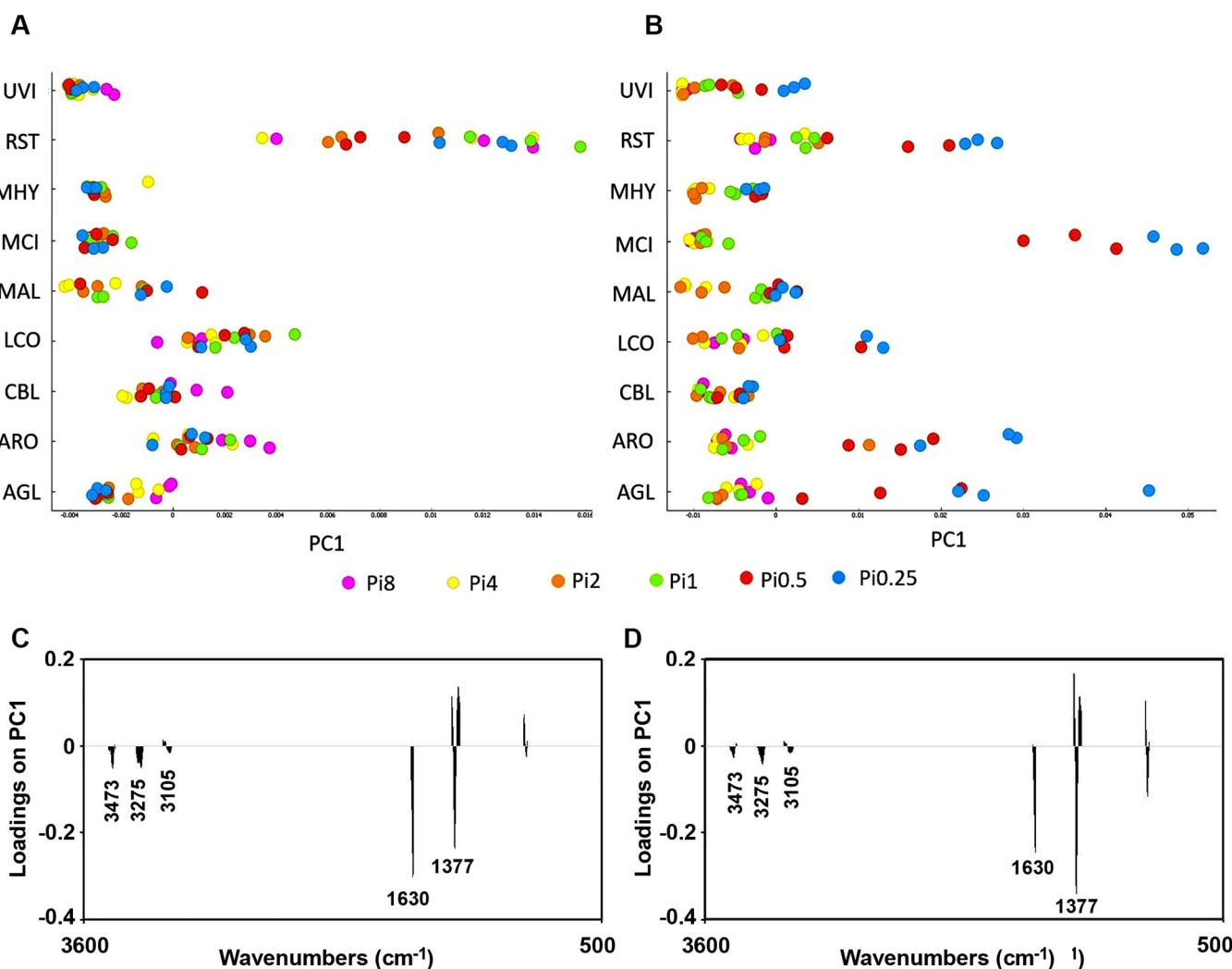

**Fig 11.** PCA results (first principal component) for the chitin/chitosan region (3457–3417 cm$^{-1}$, 3293–3251 cm$^{-1}$, 3133–3081 cm$^{-1}$, 1639–1623 cm$^{-1}$, 1392–1346 cm$^{-1}$, 962–941 cm$^{-1}$) of FTIR-HTS spectra of fungi (pre-processed by 2$^{nd}$ derivative and EMSC) grown on (A) YE-Pi and (B) AS-Pi. In A and C, the score plot and the corresponding loading plot are shown for fungi grown on YE-Pi using different Pi levels. In B and D, the score plot and the corresponding loading plot are shown for fungi grown on AS-Pi media using different Pi levels. The color coding is according to the Pi levels. The loading plots show that the total chitin/chitosan content is increasing from the left to the right in both score plots. The explained variance for the first principal component is 79% and 80% for YE and AS, respectively.

ANOVA model for the spectral region related to chitin/chitosan (3457–3417 cm$^{-1}$, 3293–3251 cm$^{-1}$, 3133–3081 cm$^{-1}$, 1639–1623 cm$^{-1}$, 1392–1346 cm$^{-1}$, 962–941 cm$^{-1}$) showed that nature of N-source may have a strong effect for *Absidia glauca* and *Mortierella hyalina*, while variation in the concentration of Pi and N-Pi interaction did not show any significant influence for these fungi (Fig 12). Generally, it could be concluded that the nature of N-source is possibly important for chitin/chitosan content for most of the studied fungi, while the influence from N-Pi interaction seemed to be least important. Variation in Pi affected chitin/chitosan content to some extend for *Amylomyces rouxii*, *Cunninghamella blakesleana*, *Mucor circinelloides* and to a high extend for *Rhizopus stolonifer* (Fig 12). Chitin/chitosan content in *Amylomyces rouxii* was equally affected by the change in N-source and Pi concentrations, while for *Rhizopus stolonifer* little effect was observed from the N-source and most of the changes where related to Pi variation.

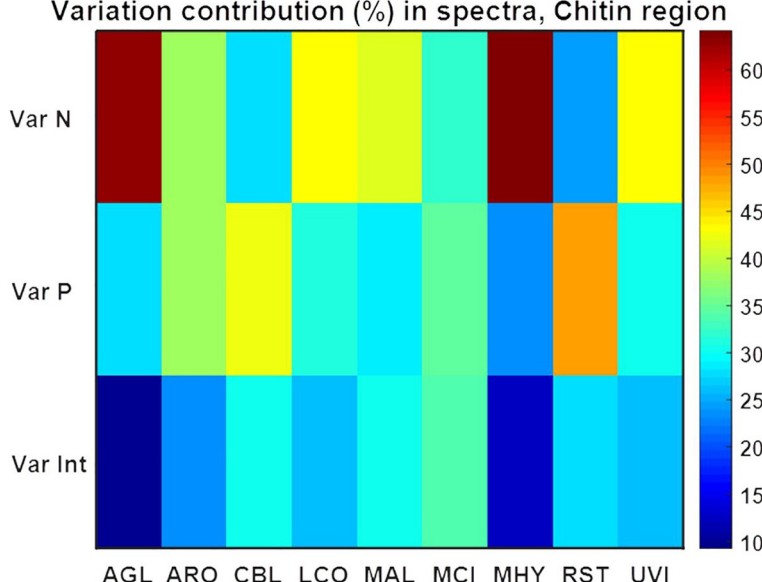

**Fig 12. Variation contribution (%) from changes in N, Pi and N-Pi interaction on the chitin/chitosan region (3457–3417 cm$^{-1}$, 3293–3251 cm$^{-1}$, 3133–3081 cm$^{-1}$, 1639–1623 cm$^{-1}$, 1392–1346 cm$^{-1}$, 962–941 cm$^{-1}$) of FTIR HTS spectra.** Variation contributions due to N and Pi alone are presented in the first two rows (Var N and Var P), whereas contribution from N-Pi interaction (Int) is presented in the last row (Var Int).

*Absidia glauca* and *Lichtheimia corymbifera* modified their cell wall mostly due to a change in nitrogen source, while variations in phosphorus changed to the chitin/chitosan production only weakly. In contrary, the variation in Pi amount induced the chitin/chitosan production in *Rhizopus stolonifer* and just small effect was observed for the contribution of different nitrogen sources. This corresponds to the biomass yield results, where there were no big changes observed with different nitrogen sources, even for the phosphorus limited conditions in the AS-Pi media. For *Amylomyces rouxii* and *Mucor circinelloides*, variation of Pi and N have similar effects on the chitin/chitosan production.

**2.2.2. Monitoring of nutrients consumption by FTIR spectroscopy.** FTIR-ATR spectra of the growth media after cultivation were used to evaluate the consumption of glucose and phosphate salts (Fig 13). We have shown recently that FTIR-ATR, in combination with multivariate statistical analyses, can be used for analysis of growth media and extracellular metabolites in screening and monitoring of fungal bioprocesses [54]. Here a univariate approach was used with only one variable per analyte (1034 and 937 cm$^{-1}$ for glucose and phosphates, respectively) in order to create a robust model for media monitoring. As shown in Fig 13, this approach allows to estimate concentrations of main nutrients in the media.

The results show that the glucose consumption corresponds to the biomass production (Fig 1). For *Mortierella alpina* and *Mortierella hyalina*, no growth was observed with the Pi8 amount, therefore the glucose content in the growth media after the cultivation was the highest. For AS-Pi media, more glucose was consumed at higher Pi concentrations (Pi1 –Pi4). For example, the double amount of biomass was produced for *Mortierella hyalina* at Pi2, compared to Pi0.25 (Fig 1), and this clearly corresponded to the lowest glucose content. For YE-Pi media, similar biomass yields were obtained irrespective of Pi concentrations, and thus the glucose consumption showed no trend. *Umbelopsis vinacea* utilized nearly all glucose available, which again corresponded to the high biomass yield.

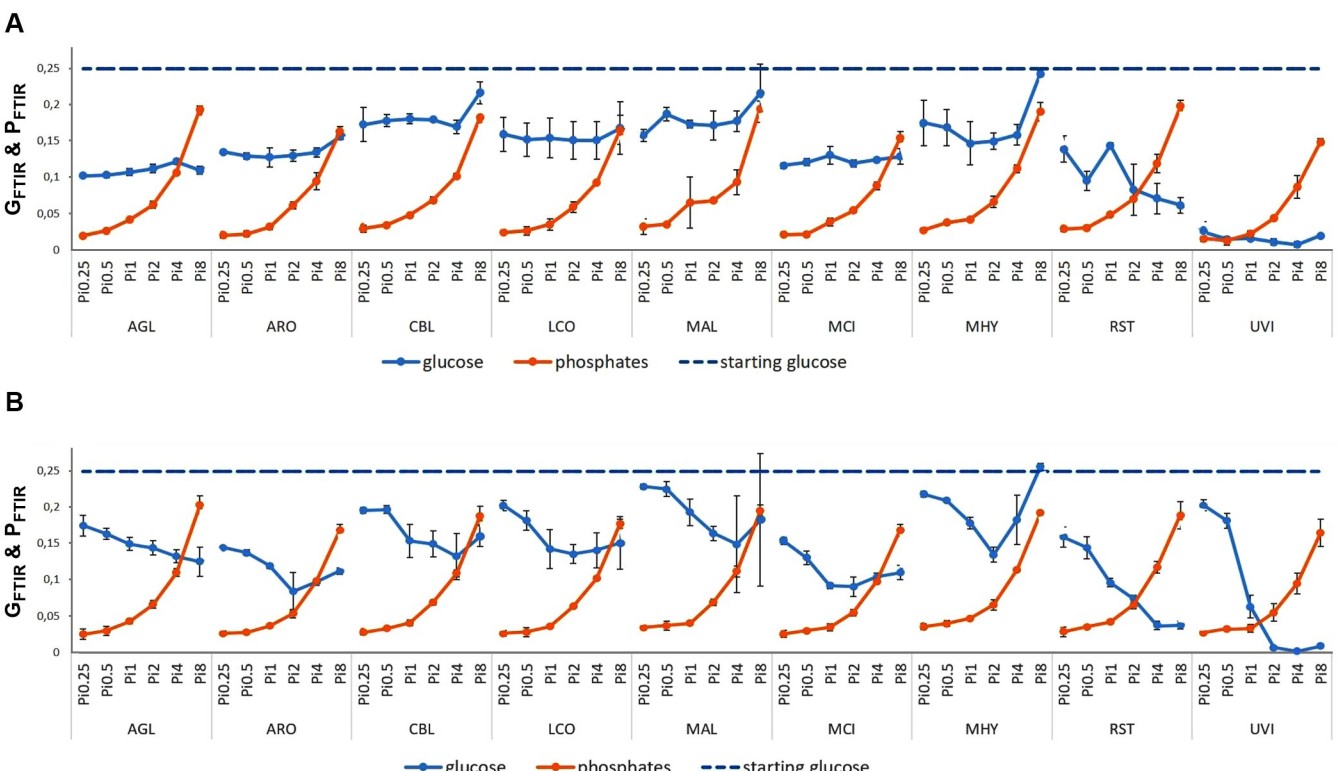

**Fig 13. Estimation of the consumption of glucose and phosphate salts in the growth media after cultivation by FTIR-ATR.** Starting glucose concentration corresponds to the blue dashed line, final glucose concentration the blue full line and final phosphates concentration are pictured with the orange line. Figure A shows the Ye-Pi media and figure B AS-Pi media.

## 3. Discussion

All studied *Mucoromycota* fungi were able to grow in media with different nitrogen sources and inorganic phosphorus concentrations, except *Mortierella* species for which the high concentration of inorganic phosphorus (Pi8) inhibited the growth completely (Fig 1). FTIR-ATR spectra of the growth media after cultivation were used to evaluate the consumption of main nutrients–glucose and phosphate salts and generally, the glucose consumption corresponds to the biomass production (Fig 1, Fig 13). In general, when yeast extract was used as a nitrogen source, no significant changes in biomass yield for different concentrations of inorganic phosphorus were observed. Moreover, biomass yield for fungi grown on yeast extract and low concentrations of inorganic phosphorus was higher compared to when ammonium sulphate was used as nitrogen source. This was mainly due to the fact, that yeast extract is a chemically complex substrate which is a source of not only nitrogen but also phosphorus, sulphur, vitamins, amino acids and trace elements. Therefore, variation in the level of inorganic phosphorus showed small effect on the fungal growth when yeast extract was used as a nitrogen source. This indicates that usage of complex N-source substrates, containing nitrogen, phosphorus and other nutrients for the enrichment of rest materials could be beneficial and sustainable. Such cultivation would provide relatively stable biomass yields without addition of inorganic phosphorus, which is a world limited chemical component. In addition, cultivation of all fungal strains in the presence of yeast extract and various inorganic phosphorus as accompanied by relative stable pH that resulted in high biomass yield.

Combining ammonium sulphate as a nitrogen source with different concentrations of inorganic phosphorus, showed that Pi- requirements for optimal growth varied for different fungi:

(1) *Rhizopus stolonifer* exhibited relatively uniform biomass yield within the used phosphorus concentration range; (2) *Mucor circinelloides*, *Absidia glauca*, *Cunninghamella blakesleeana* and *Amylomyces rouxii* showed the optimal growth and the highest biomass yield within relatively broad concentration range Pi1 –Pi8; (3) *Umbelopsis vinacea* had the highest biomass yield within more narrow concentration range Pi2 –Pi4; (4) *Lichtheimia corymbifera*, *Mortierella alpina* and *Mortierella hyalina* produced the highest biomass yield at Pi2 concentration, and, in case of *Mortierella hyalina*, the biomass yield was double compared to Pi4 and Pi1. Overall, the biomass yield for fungi grown in media with the high phosphorus levels was comparable to the yeast extract media, and in some cases, even higher biomass yields were obtained. The biomass yield clearly showed that the majority of strains, aside from *Rhizopus stolonifer*, have strongly inhibited growth in the low phosphorus media.

When the biomass yield and pH results are compared, it is apparent that low pH caused by the low Pi concentrations inhibited fungal growth. Fungal lipids are accumulated when nitrogen depletion leads to decrease of adenosine monophosphate (AMP) level. This results in the inactivation of isocitrate dehydrogenase and causes the accumulation of citric acid. Citric acid is further converted by ATP citrate lyase to AcetylCoA, which is a precursor for the synthesis of fatty acids (Wynn et al., 2001). Accumulation of citric acid, which was detected by FTIR and HPLC in our previous study (Kosa et al., 2017b), is therefore expected to some extend in the lipogenesis in oleaginous fungi. Other organic acids coming from glycolysis and the Krebs cycle, such as pyruvic or fumaric acid, might be present as well and contribute to the acidic pH. Although decrease in pH is observed in the YE-Pi media as well, the buffering capacity of YE is higher than AS (S20 Fig). Some of the fungal strains activated the protective mechanisms against acidic stress, leading to increased chitin/chitosan production in the cell wall, as it was reported above (Figs 3, 5, 6B and 11). Further studies under pH-controlled conditions are needed to clarify the contribution of low Pi to chitin/chitosan overproduction.

FTIR-HTS spectra of fungi grown on yeast extract and different Pi concentrations showed that *Mucor circinelloides*, *Umbelopsis vinacea*, *Mortierella hyalina*, *Mortierella alpina* and *Absida glauca* had high lipid content in the biomass. The highest lipid content was observed for *Umbelopsis vinacea* (Fig 6). Considering high biomass yield and lipid accumulation, *Umbelopsis vinacea* could be considered as one of the best lipid producers with the potential for industrial application. Several fungi, namely *Absidia glauca*, *Rhizopus stolonifer*, *Amylomyces rouxii*, *Mucor circinelloides*, showed accumulation of polyphosphates in addition to lipids when grown on yeast extract. *Mucor circinelloides* showed the highest content of polyphosphate co-produced along with the relatively high content of lipids (Fig 6). The co-production of lipids along with polyphosphates and chitosan by *Mucor circinelloides* could be considered as one of the co-production concepts for elevating level of sustainability for fungal lipid production, as three products would be produced in a single fermentation process.

PCA of the lipid region of FTIR-HTS spectra showed that when yeast extract was used as N-source, phosphorus availability did not affect the accumulation of lipids in *Umbelopsis vinacea*, *Mortierella hyalina*, and *Mucor circineloides*, and just minimal effect could be seen in case of *Mortierella alpina*. This indicates that for the biotechnological production of lipids by *Mucoromycota* fungi, it would be possible to exclude or limit addition of inorganic phosphorus without any strong effect on biomass and lipid yield when a rich N-source is used. This is of particular importance for reduction of costs in production of lipids for biodiesel. Accumulation of lipids by *Rhizopus stolonifer*, *Cunninghamella blakesleeana*, *Amylomyces rouxii* and *Absidia glauca* was negatively affected when a high concentration of inorganic phosphorus was present in the growth media containing either yeast extract or ammonium sulphate (Fig 7). This can be explained by the ability of co-production of lipids and polyphosphates of these fungi. Polyphosphates were accumulated during the exponential growth phase and therefore

the lipid accumulation might have been suppressed, since both, polyphosphates and lipids serve as energy storage in the fungal cells. The strongest effect of phosphorus was detected when fungi were grown on ammonium sulphate-based media, where the decrease in phosphorus availability led to a lower lipid content for all fungi except *Cunninghamella blakesleeana*. Pi2 condition doubled the biomass yield of *Mortierella* fungi known as excellent producers of high- valued polyunsaturated fatty acids.

Further, FTIR-HTS data were utilized to study nutrient-induced co-production strategies for concomitant production of lipids, polyphosphate and chitin/chitosan in *Mucoromycota* fungi. A co-production of polyphosphate and lipids was observed for fungi *Rhizopus stolonifer*, *Mucor circinelloides*, *Amylomyces rouxii* and *Absidia glauca* grown in both yeast- and ammonium-based media (Fig 6). A big increase in the polyphosphate accumulation was observed in media with the high level of phosphorus (Pi2 –Pi8). Thus, these fungi could be considered as promising strains from the view of increasing sustainability in microbial-lipid biorefinery and phosphorus recovery. The co-production of polyphosphate in addition to lipids was not detected for *Mortierella* fungi and *Umbelopsis vinacea*. The co-production of chitin/chitosan generally occured in all types of media, since these polymers are principle structural components of *Mucoromycota* fungi cell wall. The main reason for the overproduction of chitin/chitosan accompanied with the decreased lipid production in AS-Pi media under low Pi levels could be an acidic stress caused by the low Pi concentrations. The obtained results are in accordance with results reported in literature showing that chitin overproduction in the cell wall of *Mucoromycota* fungi is induced by acidic pH of the growth media [94]. Taking in account high biomass yield, the total chitin/chitosan yield in YE-Pi media could be higher than in AS-Pi media with low Pi levels. When using inorganic nitrogen source, it is possible to stimulate the overproduction of chitin/chitosan in some *Mucoromycota* fungi by limiting inorganic phosphorus. This finding is in agreement with chitosan yield optimization in *Mucor indicus*, where phosphate-free medium was reported to result in the highest chitosan production [95].

Production of chitin and chitosan from fungal mycelium has significant advantages in comparison to traditional way from crustacean waste. For example, microbial production is not dependent on a season and fishing industry, it does not require demineralization process, and the composition of chitin and chitosan is more consistent compared to crustacean waste materials [96, 97]. Chitin/chitosan creates side-stream product and additional value for the process. Although optimization of the biomass yield for chitin/chitosan and lipid co-production was not the primary goal of this study, there is a potential to enhance the yields by modification of C/N ratio, pH, aeration, cultivation temperature and time [22, 98, 99].

Generally, the co-production concept of bioproduction in some cases may lead to the use of different downstream processes, while in the case of oleaginous fungal biomass, co-produced lipids and chitin/chitosan are located in different cell compartments (lipids in lipid droplets, chitin/chitosan in cell wall) and therefore they can be separated relatively easy by using, for example, solvent-based or super critical fluid extraction. Lipids and chitin/chitosan are carbon-based products. Therefore, achieving a high yield of lipids will be at the expense of the yield of chitin/chitosan and vice versa. However, it is important to note that chitin/chitosan are the principle components of the fungal cell wall. Thus, even with the highest possible lipid yield, the cell wall, which is an essential part of the fungal cells, will always constitute a rest product after lipid extraction. The separation of polyphosphate could be challenging since its located both intracellularly and in the cell wall. Further, it's important to note that co-production concept is particularly beneficial for producing microbial biomass to be consumed as a whole, as for example fungal biomass enriched in lipids, chitin/chitosan and polyphosphates could be particularly beneficial as a whole for fish feed applications.

## 4. Conclusion

The presented study reveals a nutrient-induced co-production of industrially important metabolites, namely lipids, polyphosphates and chitin/chitosan in oleaginous *Mucoromycota* fungi using FTIR spectroscopy. The co-production was shown to depend sensitively on the presence and concentration of macronutrients in the substrates, namely six different phosphorus levels and two nitrogen sources (yeast extract and ammonium sulphate). Since the co-production of different high-value products is closely related to the sustainability of the process, our study can be considered as an assessment of the biotechnological potential of the nine different oleaginous *Mucoromycota* grown on nitrogen-limited conditions. Ammonium sulphate growth media enabled full control over the media composition, and thus the overview of the effect of different phosphorus levels on the fungal growth and metabolism.

As polyphosphate accumulating fungi, we have identified *Mucor circinelloides*, *Amylomyces rouxii*, *Rhizopus stolonifer* and *Absidia glauca*. These fungi showed a potential for the co-production of lipids and polyphosphates. Further, phosphorus limiting conditions led to low pH which induced over co-production of chitin/chitosan for *Rhizopus stolonifer*, *Mucor circinelloides*, *Amylomyces rouxii* and *Absidia glauca* in AS-Pi media. In addition, *Rhizopus stolonifer* showed an obvious advantage in managing Pi deficiency, since its growth in AS-Pi was not remarkably affected by phosphorus limitation. *Mucor circinelloides* has a high biotechnological potential for the co-production of three products, namely chitin/chitosan, lipids and polyphosphates in a single cultivation. *Umbelopsis vinacea* was identified as the best biomass and lipid producer, the yields were almost twice as high as for the other studied fungi. These findings are important for developing sustainable modern microbial lipid biorefineries. This study demonstrates that Fourier transform infrared spectroscopy allows to monitor any chemical bioprocess compound in media and cells without tedious sample preparation and extraction steps and is a powerful tool that can be used for developing and monitoring novel biotechnological processes.

## Supporting information

**S1 Fig. FTIR-HTS spectra of *Absidia glauca* (EMSC corrected); ammonium sulphate nitrogen source, different Pi-levels.**
(DOCX)

**S2 Fig. FTIR-HTS spectra of *Absidia glauca* (EMSC corrected); yeast extract nitrogen source, different Pi-levels.**
(DOCX)

**S3 Fig. FTIR-HTS spectra of *Amylomyces rouxii* (EMSC corrected); ammonium sulphate nitrogen source, different Pi-levels.**
(DOCX)

**S4 Fig. FTIR-HTS spectra of *Amylomyces rouxii* (EMSC corrected); yeast extract nitrogen source, different Pi-levels.**
(DOCX)

**S5 Fig. FTIR-HTS spectra of *Cunninghamella blakesleeana* (EMSC corrected); ammonium sulphate nitrogen source, different Pi-levels.**
(DOCX)

**S6 Fig. FTIR-HTS spectra of *Cunninghamella blakesleeana* (EMSC corrected); yeast extract nitrogen source, different Pi-levels.**
(DOCX)

**S7 Fig. FTIR-HTS spectra of *Lichtheimia corymbifera* (EMSC corrected); ammonium sulphate nitrogen source, different Pi-levels.**
(DOCX)

**S8 Fig. FTIR-HTS spectra of *Lichtheimia corymbifera* (EMSC corrected); yeast extract nitrogen source, different Pi-levels.**
(DOCX)

**S9 Fig. FTIR-HTS spectra of *Mortierella alpina* (EMSC corrected); ammonium sulphate nitrogen source, different Pi-levels.**
(DOCX)

**S10 Fig. FTIR-HTS spectra of *Mortierella alpina* (EMSC corrected); yeast extract nitrogen source, different Pi-levels.**
(DOCX)

**S11 Fig. FTIR-HTS spectra of *Mucor circinelloides* (EMSC corrected); ammonium sulphate nitrogen source, different Pi-levels.**
(DOCX)

**S12 Fig. FTIR-HTS spectra of *Mucor circinelloides* (EMSC corrected); yeast extract nitrogen source, different Pi-levels.**
(DOCX)

**S13 Fig. FTIR-HTS spectra of *Mortierella hyalina* (EMSC corrected); ammonium sulphate nitrogen source, different Pi-levels.**
(DOCX)

**S14 Fig. FTIR-HTS spectra of *Mortierella hyalina* (EMSC corrected); yeast extract nitrogen source, different Pi-levels.**
(DOCX)

**S15 Fig. FTIR-HTS spectra of *Rhizopus stolonifer* (EMSC corrected); ammonium sulphate nitrogen source, different Pi-levels.**
(DOCX)

**S16 Fig. FTIR-HTS spectra of *Rhizopus stolonifer* (EMSC corrected); yeast extract nitrogen source, different Pi-levels.**
(DOCX)

**S17 Fig. FTIR-HTS spectra of *Umbelopsis vinacea* (EMSC corrected); ammonium sulphate nitrogen source, different Pi-levels.**
(DOCX)

**S18 Fig. FTIR-HTS spectra of *Umbelopsis vinacea* (EMSC corrected); yeast extract nitrogen source, different Pi-levels.**
(DOCX)

**S19 Fig. FTIR-ATR spectra of glucose, ammonium sulphate (AS), phosphate salts, yeast extract (YE) and growth media AS-Pi4 before the cultivation.**
(DOCX)

**S20 Fig.** Titration of 100 ml not autoclaved YE-Pi0.25 (blue) and AS-Pi0.25 (red) with 1M HCl confirmed the buffering properties of yeast extract.
(DOCX)

**S1 Table. FTIR raw spectral data: HTS spectra of biomass, ATR spectra of culture supernatant, ATR spectra of reference materials, HTS spectra of culture supernatant MCI_YE_Pi1; MCI_AS_Pi1.**
(XLSX)

## Acknowledgments

We thank Lene Cecilie Hermansen from Imaging Centre at NMBU for the help with the TEM measurements.

## Author Contributions

**Conceptualization:** Simona Dzurendova, Boris Zimmermann, Achim Kohler, Volha Shapaval.

**Data curation:** Simona Dzurendova, Boris Zimmermann.

**Formal analysis:** Simona Dzurendova, Valeria Tafintseva.

**Funding acquisition:** Achim Kohler, Volha Shapaval.

**Investigation:** Simona Dzurendova, Boris Zimmermann, Ondrej Slany.

**Methodology:** Boris Zimmermann, Achim Kohler, Valeria Tafintseva, Volha Shapaval.

**Project administration:** Achim Kohler, Volha Shapaval.

**Resources:** Achim Kohler, Volha Shapaval.

**Software:** Achim Kohler.

**Supervision:** Boris Zimmermann, Achim Kohler, Volha Shapaval.

**Validation:** Achim Kohler, Valeria Tafintseva.

**Visualization:** Simona Dzurendova, Boris Zimmermann, Valeria Tafintseva.

**Writing – original draft:** Simona Dzurendova.

**Writing – review & editing:** Simona Dzurendova, Boris Zimmermann, Achim Kohler, Valeria Tafintseva, Ondrej Slany, Milan Certik, Volha Shapaval.

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
