## [Decision Letter · Decision Letter 0]

16 Apr 2020

PONE-D-20-06637

Microcultivation and FTIR spectroscopy-based screening revealed a nutrient-induced co-production of high value metabolites in oleaginous Mucoromycota fungi

PLOS ONE

Dear Simona Dzurendova,

Thank you for submitting your manuscript to PLOS ONE. After careful consideration by unfortunately a single reviewer that is a biologist and not a FTIR expert, we feel that it has merit but does not fully meet PLOS ONE’s publication criteria as it currently stands. Therefore, we invite you to submit a revised version of the manuscript that addresses the various points raised by the unique reviewer. I am actively trying to find FTIR experts to review the revised version of your manuscript.

We would appreciate receiving your revised manuscript by May 31 2020 11:59PM. To enhance the reproducibility of your results, we recommend that if applicable you deposit your laboratory protocols in protocols.io, where a protocol can be assigned its own identifier (DOI) such that it can be cited independently in the future. For instructions see: http://journals.plos.org/plosone/s/submission-guidelines#loc-laboratory-protocols

We look forward to receiving your revised manuscript.

Kind regards,

Marie-Joelle Virolle, PhD

Academic Editor

PLOS ONE

Journal Requirements:

Reviewers' comments:

Reviewer's Responses to Questions

**Comments to the Author**

1. Is the manuscript technically sound, and do the data support the conclusions?

Reviewer #1: Yes

2. Has the statistical analysis been performed appropriately and rigorously? 

Reviewer #1: Yes

3. Have the authors made all data underlying the findings in their manuscript fully available?

Reviewer #1: Yes

4. Is the manuscript presented in an intelligible fashion and written in standard English?

Reviewer #1: Yes

5. Review Comments to the Author

Reviewer #1: The authors of the article entitled "Microcultivation and FTIR spectroscopy-based screening revealed a nutrient-induced

co-production of high value metabolites in oleaginous Mucoromycota fungi" have evaluated the effect of N and P sources in the production and co-production of valuable metabolites. The article cover a high throughput study where different oleaginous microorganisms are simultaneously analysed. This manuscript represents an important milestone in the utility of FTIR spectroscopy for characterising metabolic capabilities of microbes, which may have important implications in biotechnology.

I have some comments I would like the authors to consider in order to improve the current manuscript.

- I understand the concept of bioproduction and how it can be advantageous in some cases for bioproduction, in order to increase economic viability of the processes. However, there are also important limitations that must be discussed, such as the fact that co-production would be linked most likely to as many downstream processes as products, which also increase the cost of of production and which may even cause that two compounds are difficult or impossible to separate.

-Along th same lines, I understand that the production of polyphosphate can be independent of the production of carbon based products such as lipids. However, lipids and chitins, would compete for the same kind of building blocks.

-For non-experts in the technique, it would be good to expand the differences between FTIR and Attenuated total reflectance (ATR)-FTIR. and why both are using for different purposes.

-Sentence on 321 must be rephrased.

-From the extracellular media, the authors measured glucose and phosphate and mention that other metabolites can be measured, such organic acid. What do the author observed in these cases? did they observe any other peak that could potentially correspond to the production of another valuable metabolite not considered in the study?

-The authors mention organic phosphate in YE as the reason why the addition of phosphate in that case does not affect much, which I agree. It would be good to give some numbers to this speculation, what is the usual content of phosphates in YE and how is that in relation with th Pi added.

-Several strains are used in this study. I miss an explanation of why those strains were selected and what is their known biotechnological potential.

-Can the technique provide absolute quantifications? the text is written in a very qualitative manner, just saying which strain is best on what but some kind of quantification, even if that is relative would help. Can yields and titers be quantified? if so, it would improve the manuscript a lot including some of these numbers to understand how do they correlate with previously published reports and, ultimately, the biotechnological potential. this would also allow to discuss more in detail the potential of co-production, is co-production still allowing significant levels of production when compared with the best single producers? Numbers would be especially helpful during the discussion.

-548-570. here the effect between N and P are compared. I find the wording a bit too strong, as it is difficult to truly compare what is more important, as the N experiment only compare organic vs inorganic N while the P experiment compare different concentrations of the same type of P. Different N concentrations should be measured in order to write statements like that. This can be easily solved by changing the wording.

-621. here it it discussed that phosphorus is a world limited compound and therefore it may be better to use YE. However, it must be taken into account that, eventually, you need phosphorus to make the YE, so the final balance of used P would be the same.

-In 639 the authors mention the pathway of ATP citrate lyase, which is found in most oleaginous organisms. Do all the organisms selected in this study have this activity encoded in their genome?

-Fig 2. The Y axes does not have any indication of units.

-Fig 5. Arrows should indicate what is the cell wall in each case and what are the lipid bodies.

6. PLOS authors have the option to publish the peer review history of their article (what does this mean?). If published, this will include your full peer review and any attached files.

Reviewer #1: Yes: Rodrigo Ledesma-Amaro

---

## [Author Response · Author response to Decision Letter 0]

7 May 2020

Dear Dr. Virolle,

Thank you very much for providing us the reviewer’s reports and giving us the opportunity to revise the manuscript. We have read the reviewers’ report carefully and modified the manuscript accordingly. Bellow we refer to all the changes done in the manuscript and answer all questions and comments of the reviewer. 

We would like to change the data availability statement. We provided all data with the revised manuscript- as an additional supplementary file. Therefore, we would like to state: ‘All relevant data are within the paper and its Supporting Information files.’

Review Comments to the Author

Reviewer #1 comment: The authors of the article entitled "Microcultivation and FTIR spectroscopy-based screening revealed a nutrient-induced co-production of high value metabolites in oleaginous Mucoromycota fungi" have evaluated the effect of N and P sources in the production and co-production of valuable metabolites. The article covers a high throughput study where different oleaginous microorganisms are simultaneously analyzed. This manuscript represents an important milestone in the utility of FTIR spectroscopy for characterizing metabolic capabilities of microbes, which may have important implications in biotechnology. 

I have some comments I would like the authors to consider in order to improve the current manuscript.

- I understand the concept of bioproduction and how it can be advantageous in some cases for bioproduction, in order to increase economic viability of the processes. However, there are also important limitations that must be discussed, such as the fact that co-production would be linked most likely to as many downstream processes as products, which also increases the cost of production and which may even cause that two compounds are difficult or impossible to separate.

Authors’ answer: We agree with the reviewer, that this point was not sufficiently addressed in the manuscript. According to the reviewer suggestion, we added the requested information to the end of the discussion section (see lines 751-763) in order to clarify the advantages and challenges of the co-production concept. Our reasoning in this new section is as follows.

As the reviewer is highlighting, a higher number of downstream processes will increase the cost of bioproduction and a co-production concept of bioproduction in some cases may lead to the use of different downstream processes. In the case of oleaginous fungal biomass, co-produced lipids and chitin/chitosan are located in different cell compartments (lipids in lipid droplets, chitin/chitosan in cell wall) and therefore they can be relatively easily separated by using e.g. solvent-based or supercritical fluid extraction. The separation of polyphosphate could be challenging since its located both intracellularly and in the cell wall. 

Further, we mention that a co-production concept is particularly beneficial for producing microbial biomass to be consumed as a whole, without separation and purification of individual components. For example, fungal biomass enriched in lipids, chitin/chitosan and polyphosphates could be particularly beneficial for fish feed applications. 

In addition, when developing sustainable biorefineries, it is crucial to develop flexible bioprocesses, which could be easily switched depending on the targeted products. In this case identifying microbial factories which are able performing co-production is important. 

Reviewer #1 comment: Along the same lines, I understand that the production of polyphosphate can be independent of the production of carbon-based products such as lipids. However, lipids and chitins, would compete for the same kind of building blocks.

Authors’ answer: It is true that both lipids and chitin/chitosan are carbon-based products. Therefore, achieving a high yield of lipids will be at the expense of the yield of chitin/chitosan and vice versa. However, it is important to note that chitin/chitosan are the principle components of the fungal cell wall. Thus, even if we aim at the highest possible lipid yield, the cell wall, which constitutes an essential part of the fungal cells, will always constitute a rest product after lipid extraction. We agree with the reviewer that the competition of chitin/chitosan production and lipid production with respect to carbon production was not sufficiently highlighted in the manuscript. We have therefore added a statement in lines 755-759. Moreover, as we already described in the introduction (lines 55-58), co-production of low value lipids for biofuels application and high value products such as chitin/chitosan is the only economically sustainable approach for fungal biofuels biorefineries. 

Reviewer #1 comment: For non-experts in the technique, it would be good to expand the differences between FTIR and Attenuated total reflectance (ATR)-FTIR. and why both are using for different purposes.

Authors’ answer: We expanded the section 2.2 ‘Fourier Transform Infrared spectroscopy reveals co-production in oleaginous Mucoromycota fungi’ related to FTIR-HTS and FTIR-ATR in lines 371-381 by adding the following paragraph. “In infrared spectroscopy, the loss of infrared radiation due to chemical absorption is quantified. In the FTIR-HTS transmission mode, the loss of radiation due to absorption is quantified by transmitting infrared radiation through a sample and quantifying the loss of the radiation by comparing the transmitted radiation with the radiation that impinges on the sample. By covering the complete spectra range of the mid-infrared, biochemical fingerprint of all major chemical building blocks is obtained. The FTIR-HTS system employs a high-throughput setup with microplates and automated measurements allowing the automated analysis of around 180 samples in one measurement run. Relatively large variance in sample thickness results in the difference in optical path length, which can be corrected by standard pre-processing tools developed by us (Kohler et al, 2005; Zimmermann et al, 2013). In FTIR-ATR analysis, the infrared radiation undergoes reflection in an ATR crystal an produces an evanescent field in the sample which is located on its surface. The evanescent field is attenuated by the sample die to chemical absorption and the absorption can be quantified by relating the attenuated radiation with the radiation that is obtained in an ATR setup without a sample at the surface of the crystal. The ATR setup is characterized by a high reproducibility caused by a stable penetration depth of the IR beam into the sample, when the sample at the top of the crystal is in tight contact with the surface of the crystal. This is true for liquid and viscous samples such as the culture supernatant in our measurements.”

Reviewer #1 comment: Sentence on 321 (324 in revised manuscript) must be rephrased.

Authors’ answer: The sentence was split and rephrased according to the reviewer suggestion (see line 324-332): "Figure 1A shows the effect of yeast extract, a complex organic multi-component substrate containing both nitrogen and phosphorus, on the cultivation of Mucoromycota fungi under different Pi levels. The results indicate that the addition of inorganic phosphorus could be neglected, since it does not have any significant effect on the biomass production.”

Reviewer #1 comment: From the extracellular media, the authors measured glucose and phosphate and mention that other metabolites can be measured, such organic acid. What do the author observed in these cases? did they observe any other peak that could potentially correspond to the production of another valuable metabolite not considered in the study?

Authors’ answer: 

Organic acids can be precisely detected by FTIR spectroscopy, as we have shown in our previous study (Kosa et al, 2017). In some cases, we observed a peak associated with a C=O stretch at 1725 cm-1, corresponding to the organic acids on the FTIR-HTS spectra of culture supernatants. However, we decided not to include data of culture supernatants in this manuscript as it would go beyond the scope of the manuscript. We did not observe any other peak in the FTIR-ATR spectra which could correspond to an additional high-value metabolite in the extracellular media.

Reviewer #1 comment: The authors mention organic phosphate in YE as the reason why the addition of phosphate in that case does not affect much, which I agree. It would be good to give some numbers to this speculation, what is the usual content of phosphates in YE and how is that in relation with the Pi added.

Authors’ answer: We added information about the P-content in YE to the manuscript, see lines 332 - 334: "Yeast extract contains approximately 2.5% of total phosphorus. This amount corresponds to approx. 15% in terms of total P contained in added phosphates salts in the lowest examined Pi condition- Pi0.25." 

Reviewer #1 comment: Several strains are used in this study. I miss an explanation of why those strains were selected and what is their known biotechnological potential.

Authors’ answer: The selection and the biotechnological potential of selected strains is shortly described in lines 145 - 150: "The selection of fungal strains was based on the results of our recent study, where 100 oleaginous filamentous fungi were screened for their ability to accumulate high amount of lipids. While some Mucoromycota species have been previously identified as medically important, in general they have been utilised at industrial scale as cell factories for example for chitosan, lipids or lactic acid production. " In addition, in the introduction we have provided overview of studies and several examples of Mucoromycota products, co-production strategies and possible substrates which they were able to utilize (see lines 41-44, 58-62). We hope this explanation is sufficient.

Reviewer #1 comment: Can the technique provide absolute quantifications? the text is written in a very qualitative manner, just saying which strain is best on what but some kind of quantification, even if that is relative would help. Can yields and titers be quantified? if so, it would improve the manuscript a lot including some of these numbers to understand how do they correlate with previously published reports and, ultimately, the biotechnological potential. this would also allow to discuss more in detail the potential of co-production, is co-production still allowing significant levels of production when compared with the best single producers? Numbers would be especially helpful during the discussion.

Authors’ answer: We clarified this aspect in following paragraph, which was added in lines 403-414: “Fourier transform infrared (FTIR) spectroscopy can provide both qualitative and quantitative measures. Quantitative analysis by FTIR requires regression onto reference data. For regression analysis often methods based on latent variables such as partial least square regression are used. As reference data for respective metabolites, e.g. chromatography analyses can be used. Qualitative measures are achieved by spectral assignments (see Figure 3 and Table 3) and by applying unsupervised multivariate data analysis tools (for example principal component analysis or ANOVA-PCA). Although FTIR spectroscopy cannot provide absolute quantifications without establishing calibration models based on reference quantitative data, a semi-quantitative analysis of ratios of chemical constituents (see Figure 13) can be obtained. Nevertheless, the biggest advantage of the FTIR approach is that it allows high-throughput screening of samples and detection of a vast range of different metabolites simultaneously within a single analytical run. Thus, it provides high precision qualitative information allowing to pre-select strains and growth conditions. “

The main aim of the study was to perform a high-throughput screening for testing many different growth conditions and fungal strains (324 cultivations were performed in total) for: 1) identifying the most promising strains and growth conditions for the co-production, and for 2) understanding the influence of N and P on the co-production of lipids, chitin/chitosan and polyphosphates. For the high-throughput screening we used microcultivation in microtiter plates that provides a relatively small amount of biomass which was not sufficient for the reference analyses of several metabolites. Therefore, we utilized FTIR spectroscopy as rapid qualitative technique providing multi-analyte information on a very high number of samples (approx. 2000 biomass samples and media samples were measured in total). 

Reviewer #1 comment: Line 548-570 (lines 605-611 in the revised manuscript), here the effect between N and P are compared. I find the wording a bit too strong, as it is difficult to truly compare what is more important, as the N experiment only compare organic vs inorganic N while the P experiment compare different concentrations of the same type of P. Different N concentrations should be measured in order to write statements like that. This can be easily solved by changing the wording.

Authors’ answer: We modified the wording according to the reviewer suggestion: "ANOVA model for the spectral region related to chitin/chitosan (3457 – 3417 cm-1, 3293 – 3251 cm-1, 3133 – 3081 cm-1, 1639 – 1623 cm-1, 1392 – 1346 cm-1, 962 – 941 cm-1) showed that nature of N-source may have a strong effect for Absidia glauca and Mortierella hyalina, while variation in the concentration of Pi and N-Pi interaction did not show any significant influence for these fungi (Fig. 12). Generally, it could be concluded that the nature of N-source is possibly important for chitin/chitosan content for most of the studied fungi, while the influence from N-Pi interaction seemed to be least important. " -line 605-611.

Reviewer #1 comment: Line 621 (line 661 in the revised manuscript), here it is discussed that phosphorus is a world limited compound and therefore it may be better to use YE. However, it must be taken into account that, eventually, you need phosphorus to make the YE, so the final balance of used P would be the same.

Authors’ answer: It is true that when using yeast extract, the final P balance would be the same. However, our statements concerned the general use of complex N-source substrates, which could be enriched with organic phosphorus. By utilizing such substrates, we could recover phosphorus in the form of cellular polyphosphates, which would be highly beneficial taking in account the global phosphorus limitations. In some cases, we would need to enrich substrates with additional nitrogen and phosphorus and in this case yeast extract is a good option since it contains both components and also other beneficial components in comparison to the simple inorganic N source such as ammonium sulphate. Based on our results, we conclude that if such complex organic materials as YE are used as substrates, there is no need to add inorganic phosphorus. We agree with the reviewer that this point was not clear in the manuscript and modified the sentence in lines 661-662 in order to make the point clearer: ’This indicates that usage of complex N-source substrates, containing nitrogen, phosphorus and other nutrients for the enrichment of rest materials could be beneficial and sustainable. Such cultivation would provide relatively stable biomass yields without addition of inorganic phosphorus, which is a world limited chemical component.’

Reviewer #1 comment: In 639 (line 685 the revised manuscript) the authors mention the pathway of ATP citrate lyase, which is found in most oleaginous organisms. Do all the organisms selected in this study have this activity encoded in their genome?

Authors’ answer: All fungal strains used in the study are oleaginous and able to accumulate more than 20% of lipids, while we did not check the ATP citrate lyase activity. The selection of fungal strains is explained in the paragraph ‘Oleaginous filamentous fungi’ (see lines 145 – 151). Strains used in the study were selected based on our previous screening of hundred oleaginous fungi for the high- and low value lipid production. 

Reviewer #1 comment: Fig 2. The Y axes does not have any indication of units.

Authors’ answer: Y axes captions were added in Fig 2.

Reviewer #1 comment: Fig 5. Arrows should indicate what is the cell wall in each case and what are the lipid bodies.

Authors’ answer: Arrows indicating cell wall and lipid bodies were added in Fig 5 (and in the figure caption).

---

## [Decision Letter · Decision Letter 1]

4 Jun 2020

Microcultivation and FTIR spectroscopy-based screening revealed a nutrient-induced co-production of high value metabolites in oleaginous Mucoromycota fungi

PONE-D-20-06637R1

Dear Dr. Simona Dzurendova,

We’re pleased to inform you that your manuscript has been judged scientifically suitable for publication and will be formally accepted for publication once it meets all outstanding technical requirements.

Kind regards,

Marie-Joelle Virolle, PhD

Academic Editor

PLOS ONE

Additional Editor Comments (optional):

Reviewers' comments:

Reviewer's Responses to Questions

**Comments to the Author**

1. If the authors have adequately addressed your comments raised in a previous round of review and you feel that this manuscript is now acceptable for publication, you may indicate that here to bypass the “Comments to the Author” section, enter your conflict of interest statement in the “Confidential to Editor” section, and submit your "Accept" recommendation.

Reviewer #1: All comments have been addressed

Reviewer #2: All comments have been addressed

2. Is the manuscript technically sound, and do the data support the conclusions?

Reviewer #1: Yes

Reviewer #2: Yes

3. Has the statistical analysis been performed appropriately and rigorously? 

Reviewer #1: Yes

Reviewer #2: Yes

4. Have the authors made all data underlying the findings in their manuscript fully available?

Reviewer #1: Yes

Reviewer #2: Yes

5. Is the manuscript presented in an intelligible fashion and written in standard English?

Reviewer #1: Yes

Reviewer #2: Yes

6. Review Comments to the Author

Reviewer #1: The authors addressed all my comments and significantly improved the manuscipt. I suggest the acceptance of the work.

Reviewer #2: I have read carefully the revised manuscript PONE-D-20-06637_R1 entitled “Microcultivation and FTIR spectroscopy-based screening revealed a nutrient-induced co-production of high value metabolites in oleaginous Mucoromycota fungi ”. The authors of this work present here a paper they wrote about the relative quantification of various metabolites in diverse media using FTIR HTS or ATR approach.

All the answers given to the previous reviewers are really satisfactory. So, my principal remark will concern the main substance of this article that clearly bring novelty and great interest to the community regarding the current state of the art. I would recommend this manuscript for publication in the PLOS journal in its actual revised form.

Sincerely yours.

7. PLOS authors have the option to publish the peer review history of their article (what does this mean?). If published, this will include your full peer review and any attached files.

Reviewer #1: Yes: Rodrigo Ledesma-Amaro

Reviewer #2: No

---

## [Editor Report · Acceptance letter]

10 Jun 2020

PONE-D-20-06637R1 

Microcultivation and FTIR spectroscopy-based screening revealed a nutrient-induced co-production of high value metabolites in oleaginous Mucoromycota fungi 

Dear Dr. Dzurendova:

I'm pleased to inform you that your manuscript has been deemed suitable for publication in PLOS ONE. Congratulations! Your manuscript is now with our production department. 

Kind regards, 

on behalf of

Dr. Marie-Joelle Virolle 

Academic Editor

PLOS ONE